# UNDERSTANDING SHARPNESS-AWARE MINIMIZATION

## ABSTRACT

Sharpness-Aware Minimization (SAM) is a recent training method that relies on worst-case weight perturbations. SAM significantly improves generalization in various settings, however, existing justifications for its success do not seem conclusive. First, we analyze the implicit bias of SAM over diagonal linear networks, and prove that it always chooses a solution that enjoys better generalisation properties than standard gradient descent. We also provide a convergence proof of SAM for non-convex objectives when used with stochastic gradients and empirically discuss the convergence and generalization behavior of SAM for deep networks. Next, we discuss why SAM can be helpful in the noisy label setting where we first show that it can help to improve generalization even for linear classifiers. Then we discuss a *gradient reweighting interpretation* of SAM and show a further beneficial effect of combining SAM with a robust loss. Finally, we draw parallels between overfitting observed in learning with noisy labels and in adversarial training where SAM also improves generalization. This connection suggests that, more generally, techniques from the noisy label literature can be useful to improve robust generalization.

## 1 INTRODUCTION

Understanding generalization of overparametrized deep neural networks is a central topic of the current machine learning research. Their training objective has many global optima where the training data are perfectly fitted (Zhang et al., 2016), but different global optima lead to dramatically different generalization performance (Liu et al., 2019). However, it has been observed that stochastic gradient descent (SGD) tends to converge to well-generalizing solutions, even *without* any explicit regularization methods (Zhang et al., 2016). This suggests that the leading role is played by the *implicit* bias of the optimization algorithms used (Neyshabur et al., 2014): when the training objective is minimized using a particular algorithm and initialization method, it converges to a specific solution with favorable generalization properties. However, even though SGD has a very beneficial implicit bias, significant overfitting can still occur, particularly in the presence of label noise (Nakkiran et al., 2019) and adversarial perturbations (Rice et al., 2020).

Recently it has been observed that the *sharpness* of the training loss, i.e., how quickly it changes in some neighborhood around the parameters of the model, correlates well with the generalization error (Keskar et al., 2016; Jiang et al., 2019), and generalization bounds related to the sharpness have been derived (Dziugaite & Roy, 2018). The idea of minimizing the sharpness to improve generalization has motivated recent works of Foret et al. (2021) and Wu et al. (2020) which propose to use worst-case perturbations of the weights on every iteration of training in order to improve generalization. We refer to this method as *Sharpness-Aware Minimization* (SAM) and focus mainly on the version proposed in Foret et al. (2021) that performs only one step of gradient ascent to approximately solve the weight perturbation problem before updating the weights.

Despite the fact that SAM significantly improves generalization in various settings, the existing justifications based on the generalization bounds provided by Foret et al. (2021) and Wu et al. (2020) do not seem conclusive. The main reason is that their generalization bounds do not distinguish the *worst-case* robustness to worst-case weight perturbation from *average-case* robustness to Gaussian noise. However the latter does not sufficiently improve generalization as both Foret et al. (2021) and Wu et al. (2020) report. Moreover, their analysis does not distinguish whether the weight perturbation is applied based on some or on all training examples which, as we will discuss, has an important impact on generalization.

We aim at further investigating the reasons for SAM's success and make the following contributions:

- We first discuss that the SAM formulation can lead to two different objectives. We show that the gradient flows on these objectives over diagonal linear networks are implicitly biased towards solutions which enjoy better generalization properties compared to the standard gradient flow.

- We provide convergence results for the SAM algorithm used with stochastic gradients for non-convex objectives and discuss the convergence and generalization behavior of SAM for deep networks.

- We discuss why SAM can prevent overfitting in the noisy label setting by interpreting SAM as a *gradient reweighting scheme* and showing the further benefits of combining it with a robust loss.

- Finally, we draw parallels between overfitting in learning with noisy labels and in adversarial training where SAM also improves generalization. This connection suggests that techniques from the noisy label literature can be more generally useful to improve robust generalization.

## 2 RELATED WORK

Here we discuss relevant works on robustness in the *weight space* and cover the main references on overfitting in the noisy label setting and in adversarial training, two settings where weight-space robustness also improves generalization.

**Weight-space robustness.** Works on weight-space robustness of neural networks date back at least to 1990s (Murray & Edwards, 1993; Hochreiter & Schmidhuber, 1995). Random perturbations of the weights are used extensively in deep learning (Jim et al., 1996; Graves et al., 2013), and most prominently in approaches such as dropout (Srivastava et al., 2014). Many practitioners have observed that using SGD with larger batches for training leads to worse generalization (LeCun et al., 2012), and Keskar et al. (2016) have shown that this degradation of performance is *correlated* with the sharpness of the found parameters. This observation has motivated many further works which focus on closing the generalization gap between small-batch and large-batch SGD (Wen et al., 2018; Haruki et al., 2019; Lin et al., 2020). More recently, Jiang et al. (2019) have shown a strong correlation between the sharpness and the generalization error on a large set of models under a variety of different settings hyperparameters, beyond the batch size. This has motivated the idea of minimizing the sharpness during training to improve standard generalization, leading to Sharpness-Aware Minimization (SAM) (Foret et al., 2021). SAM modifies SGD such that on every iteration of training, the gradient is taken not at the current iterate but rather at a worst-case point in its vicinity. Zheng et al. (2021) concurrently propose a very similar weight perturbation method which also successfully improves standard generalization on multiple deep learning benchmarks. Wu et al. (2020) have also proposed a similar algorithm with the same motivation, although Wu et al. (2020) focuses on improving robust generalization of adversarial training.

**Overfitting in learning with noisy labels.** Arpit et al. (2017) have observed that deep networks noticeably overfit in the presence of mislabeled samples but early stopping can mitigate the problem. There are multiple other approaches that can mitigate this overfitting behavior: removing (Song et al., 2020) or downweighting noisy points (Jiang et al., 2018; Huang et al., 2020), using robust losses (Ghosh et al., 2017; Zhang & Sabuncu, 2018; Menon et al., 2019). Most of these approaches explicitly or implicitly leverage the *early-learning phenomenon* (Liu et al., 2020) where the network tends to fit first correctly labeled points and noisy points later in training. In the context of adversarial robustness, Sanyal et al. (2020) have discussed that fitting label noise can aggravate adversarial vulnerability, although removing label noise is not sufficient to achieve adversarial robustness.

**Overfitting in adversarial training.** Adversarial training in deep learning has been formulated as a robust optimization problem by Madry et al. (2018) where the worst-case perturbations are typically found via projected gradient descent. Rice et al. (2020) have described the *robust overfitting* phenomenon in adversarial training suggesting that early stopping is highly beneficial and provides a competitive baseline. However, they do not provide explanations about the reasons behind the overfitting phenomenon. The state-of-the-art approaches propose different ways to improve robust generalization and mitigate robust overfitting such as using additional training data (Carmon et al., 2019), weight averaging (Gowal et al., 2020; Chen et al., 2021), or advanced data augmentations (Rebuffi et al., 2021). Additionally, Wu et al. (2020) have shown that combining adversarial training with a method similar to sharpness-aware minimization also improves robust generalization.

# 3 IMPLICIT BIAS AND CONVERGENCE OF SAM

In this section, we discuss different objectives related to Sharpness-Aware Minimization (SAM) and their generalization properties, then derive convergence results for the stochastic SAM algorithm, and further discuss the convergence and generalization of SAM for deep networks.

## 3.1 IMPLICIT BIAS OF SAM

**SAM objectives.** Let $\{x_i, y_i\}_{i=1}^n$ be the training data points and $\ell_i(w)$ be the loss of a classifier parametrized by weights $w \in \mathbb{R}^{|w|}$ and evaluated at point $(x_i, y_i)$. Foret et al. (2021) theoretically base their approach on the following objective which we denote as *MaxSum SAM*:

$$\textbf{MaxSum SAM:} \quad \min_{w \in \mathbb{R}^{|w|}} \max_{\|\delta\|_2 \le \rho} \frac{1}{n} \sum_{i=1}^n \ell_i(w + \delta). \tag{1}$$

They justify this objective via a PAC-Bayesian generalization bound, although they show empirically (see Fig. 3 therein) that the following objective, denoted as *SumMax SAM*, leads to better generalization:

$$\textbf{SumMax SAM:} \quad \min_{w \in \mathbb{R}^{|w|}} \frac{1}{n} \sum_{i=1}^n \max_{\|\delta\|_2 \le \rho} \ell_i(w + \delta). \tag{2}$$

The update rule of SAM for these objectives amounts then to a variation of gradient descent with step size $\gamma_t$ where the gradients are taken at intermediate points $w_{t+1/2}^i$, i.e., $w_{t+1} = w_t - \frac{\gamma_t}{n} \sum_{i=1}^n \nabla \ell_i(w_{t+1/2}^i)$. The two objectives, however, differ in how the points $w_{t+1/2}^i$ are computed since they approximately maximize different losses with inner step sizes $\rho_t$:

$$\textbf{MaxSum:} \ w_{t+1/2}^i = w_t + \frac{\rho_t}{n} \sum_{j=1}^n \nabla \ell_j(w_t) \quad \text{vs.} \quad \textbf{SumMax:} \ w_{t+1/2}^i = w_t + \rho_t \nabla \ell_i(w_t). \tag{3}$$

We next show formally that SumMax SAM has a better implicit bias than ERM and MaxSum SAM.

**Implicit bias of SumMax and MaxSum SAM.** The implicit bias of gradient methods is well understood for linear models where all gradient-based algorithms enjoy the same implicit bias. For diagonal linear neural networks, first-order algorithms have a richer implicit bias. We consider here a sparse regression problem and a predictor parametrized as $f_{u,v} = u \odot u - v \odot v$. Before understanding how SAM induces a preferable bias, we first recall the seminal result of Woodworth et al. (2020): assuming global convergence, the solution selected by the gradient flow initialised at $\alpha \in \mathbb{R}^d$ and denoted $\beta_\infty^\alpha$ solves the constrained optimisation problem:

$$\beta_\infty^\alpha = \underset{\beta \in \mathbb{R}^d \text{ s.t. } X\beta = y}{\arg \min} \phi_\alpha(\beta). \tag{4}$$

The potential $\phi_\alpha$ whose precise expression is given Eq. (12) in App. A interpolates nicely between the $\ell_1$ and the $\ell_2$ norms according to the initialization scale $\alpha$: Small initialisations lead to low $\ell_1$-type solutions which are known to induce good generalisation properties. Large initialisations lead to low $\ell_2$-type solutions. Our main result is that both MaxSum and SumMax dynamics bias the flow towards solutions which still minimise the potential $\phi_\alpha$. However it does so with effective parameter $\alpha_{\text{MaxSum}}$ and $\alpha_{\text{SumMax}}$ which are strictly smaller than $\alpha$ for suitable inner step size $\rho$. Thus the chosen solution has better sparsity-inducing properties than the solution of the gradient flow.

**Proposition 1** (Informal). *Assuming global convergence, the solutions selected by the MaxSum and the SumMax algorithms defined Eq. (3), taken in the infinitesimally-small-$\gamma_t$ limit and initialised at $\alpha$, solve the optimisation problem (4) with effective parameters $\alpha_{SumMax}$ and $\alpha_{SumMax}$ which satisfy:*

$$\alpha_{SumMax} = \alpha e^{-\rho \Delta_{SumMax} + O(\rho^2)} \text{ and } \alpha_{MaxSum} = \alpha e^{-\rho \Delta_{MaxSum} + O(\rho^2)},$$

*where $\Delta_{SumMax}$ and $\Delta_{MaxSum}$ are two entrywise positive vectors for which typically:*

$$\|\Delta_{SumMax}\|_1 \approx d \int_0^\infty L(w(s)) ds \text{ and } \|\Delta_{MaxSum}\|_1 \approx \frac{d}{n} \int_0^\infty L(w(s)) ds.$$

The results are formally stated in Proposition 4 and 5 in App. A. The SumMax implementations has better bias properties since its effective scale of $\alpha$ is considerably smaller than the one of MaxSum. It is worth noting that the vectors $\Delta_{\text{SumMax}}$ and $\Delta_{\text{MaxSum}}$ are linked with the integral of the loss function along the flow. Thereby, the speed of convergence of the training loss impacts the magnitude of the biasing effect: the slower the convergence, the better the bias, similarly to what observed for SGD by Pesme et al. (2021).

**Empirical evidence for the implicit bias.** We compare the performance of different methods (ERM, MaxSum and SumMax SAM) on the training and test losses in Fig. 1. As predicted, the methods enjoy various generalization abilities: ERM and MaxSum SAM enjoy the same performance whereas SumMax benefits from a better implicit bias. We also note that the training loss of all the variants is converging to zero but as alluded before the convergence of SumMax SAM is slower. We show a similar experiments in App. A with stochastic variants of the algorithms. As expected, their performances are better than their deterministic counterparts (Keskar et al., 2016; Pesme et al., 2021).

The results on the implicit bias presented above require that the algorithm converges to zero training error. Therefore we analyze next the convergence of the SAM algorithm for general non-convex functions in the practically relevant stochastic case. Note that we cannot expect rates as fast as the one observed for the previous simple model. They were due to its special structure for which fast convergence rates have been proven (Yun et al., 2021).

**Figure 1:** Implicit bias of Sum-Max SAM for a diagonal network on a sparse regression problem.

### 3.2 Convergence of SAM algorithm

To make the SAM algorithm practical, Foret et al. (2021) propose to combine SAM with stochastic gradients. By denoting the batch indices at time $t$ by $I_t$, this leads to the following update rule:

$$w_{t+1} = w_t - \frac{\gamma_t}{|I_t|} \sum_{i \in I_t} \nabla \ell_i \big( w_t + \frac{\rho_t}{|I_t|} \sum_{j \in I_t} \nabla \ell_j(w_t) \big). \tag{5}$$

Importantly, the *same* batch $I_t$ is used for the inner and outer gradient steps which allows to interpolate between the MaxSum (for $|I_t| = n$) and SumMax (for $|I_t| = 1$) update rules defined in Eq. (3). However, using batch size $|I_t| = 1$ is inefficient in practice since it does not fully utilize the computational accelerators such as GPUs so for most experiments, Foret et al. (2021) use a higher $|I_t|$ to balance the generalization improvement with the computational efficiency.

In the following, we analyze the convergence of the update rule from Eq. (5). We make the following assumptions on the training loss $L(w) = \frac{1}{n} \sum_{i=1}^{n} \ell_i(w)$:

**(A1)** (Bounded variance). *It exists $\sigma \geq 0$ s.t. $\mathbb{E}[\|\nabla \ell_i(w) - \nabla L(w)\|^2] \leq \sigma^2, i \sim \mathcal{U}(\llbracket 1, n \rrbracket), w \in \mathbb{R}^d$.*

**(A2)** (Individual $\beta$-smoothness). *It exists $\beta > 0$ s.t. $\|\nabla \ell_i(w) - \nabla \ell_i(v)\| \leq \beta \|w - v\|$ for all $w, v \in \mathbb{R}^d$ and $i \in \llbracket 1, n \rrbracket$.*

**(A3)** (Polyak-Lojasiewicz). *It exists $\mu > 0$ s.t. $\frac{1}{2}\|\nabla L(w)\|^2 \geq \mu(L(w) - L_*)$ for all $w, v \in \mathbb{R}^d$.*

We have the following convergence result for the stochastic SAM algorithm (see proof in App B.2).

**Proposition 2.** *Assume (A1-2) for the iterates (5). For any $T \geq 0$ and for step-sizes $\gamma_t = \frac{1}{\sqrt{T}\beta}$ and $\rho_t = \frac{1}{T^{1/4}\beta}$, we have:*

$$\frac{1}{T} \mathbb{E} \Big[ \sum_{t=0}^{T-1} \|\nabla L(w_t)\|^2 \Big] \leq \frac{4}{\beta\sqrt{T}}(L(w_0) - L_*) + \frac{8\sigma^2}{b\sqrt{T}},$$

*In addition, under (A3), with step-sizes $\gamma_t = \min\{\frac{8t+4}{3\mu(t+1)^2}, \frac{1}{2\beta}\}$ and $\rho_t = \sqrt{\gamma_t/\beta}$:*

$$\mathbb{E}[L(w_T)] - L_* \leq \frac{3\beta^2(L(w_0) - L_*)}{\mu^2 T^2} + \frac{22\beta\sigma^2}{\mu^2 bT}$$

We make several remarks regarding this theorem:

- We recover the rates of SGD with the usual condition on the step size $\gamma_t$ (Ghadimi & Lan, 2013; Karimi et al., 2016). The ascent step-size $\rho_t$ has to be $O(\sqrt{\gamma_t})$ to ensure convergence. Therefore it tolerates a slower decrease than $\gamma_t$. This finding is aligned with the fact that the ascent step-size of SAM should not be decreased as drastically as the descent step-size when training neural networks.
- The proof relies on the bound $\langle \nabla L(w_t + \eta \nabla L(w_t)), \nabla L(w_t) \rangle \geq (1 - \eta\beta)\|\nabla L(w_t)\|^2$ which shows that SAM-step is well aligned with the gradient step (seem Lemma 16 in App B.2).
- We can obtain the same convergence result for the different variants of SAM described in the previous section. We also provide in App B.1 tighter convergence rates for the full-batch case.

**SAM with normalized gradient.** When SAM algorithm was first introduced, the gradient was normalized in the inner step, in order to have an ascent-step of size exactly $\rho$. This normalization is not needed to get similar performance in applications and is not favorable for optimization purposes. Indeed when used with constant step-size normalized SAM is oscillating, and the averaged iterate converges to a different point which is not necessary flatter than the solution. When decreasing step-sizes are used, then the behaviors are comparable to the ones in the unnormalized case. In App. B.3, we show experiments that suggest that SAM with and without normalization steps achieves similar improvements in terms of generalization. However, for all other experiments in the paper, we rely on the *normalized* SAM as introduced in Foret et al. (2021) which is the main focus of our study.

### 3.3 Convergence and generalization of SAM for deep networks

Here we relate the theoretical results from the previous section with the behavior of SAM for deep networks on a standard image classification dataset.

**ERM and SAM both converge but generalize differently.** We compare the behavior of ERM and SAM by training a ResNet-18 on CIFAR-10 for 1000 epochs using mini-batch SGD with momentum and piece-wise constant learning rates (see App. C for experimental details) and plot the results over epochs in Fig. 2. We observe that not only the ERM model but also the model trained with SAM fits all the training points and converges to a *nearly zero training loss*: 0.0012 for ERM vs 0.0009 for SAM. However, the SAM model has significantly better generalization performance: 3.76% vs 5.03% test error. Moreover, we observe no noticeable overfitting on standard CIFAR-10: the best and last model differ at most by 0.1% test error for both methods.

**When is implicit bias of SAM beneficial during training?** To better understand the implicit bias of SAM over training iterations, we perform the following experiment: we train with ERM for 900 epochs and then use SAM for the remaining 100 epochs (ERM → SAM), and vice versa (SAM → ERM). We can see from Fig. 2 (right) that for SAM → ERM once SAM converges to a well-generalizing minimum thanks to its implicit bias, then it is not important whether we continue optimization with SAM or with ERM.

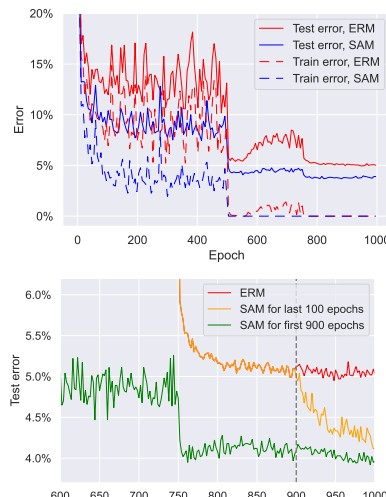

**Figure 2:** ERM vs SAM training over epochs. **Top**: standard SAM training. **Bottom**: SAM training enabled for the first 900 epochs and last 100 epochs.

In particular, we do not observe overfitting when switching to ERM. We note that it is still important here that SAM converges since SAM has to minimize the training loss below some threshold after which ERM can be potentially applied.

At the same time, for ERM → SAM we observe a different behavior: the test error clearly improves when switching from ERM to SAM. This suggests that SAM (using a higher $\rho$ than the standard value, see App. C) can help to escape a suboptimal minimum where ERM converges to. This phenomenon is interesting since it suggests that such a fine-tuning scheme can save computations as we can potentially start from any pre-trained model and improve its generalization on the *same* dataset which is different from fine-tuning with SAM in the transfer learning setup covered in Foret et al. (2021).

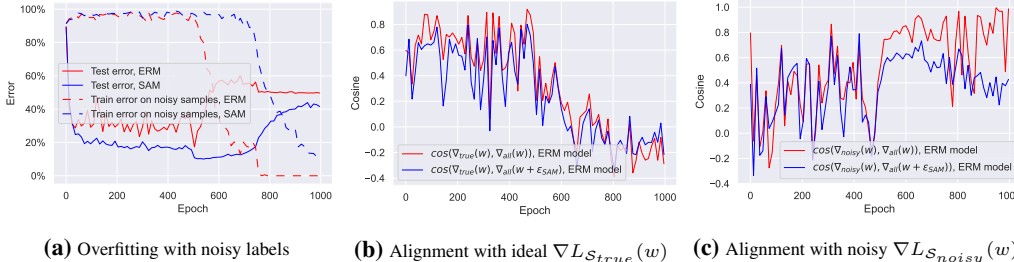

**(a)** Overfitting with noisy labels    **(b)** Alignment with ideal $\nabla L_{\mathcal{S}_{true}}(w)$    **(c)** Alignment with noisy $\nabla L_{\mathcal{S}_{noisy}}(w)$

**Figure 3:** Plots over training for a ResNet-18 trained on CIFAR-10 with 60% label noise. We can observe that (a) test error increases when we fit the noisy samples, (b) using gradients at the SAM point doesn't change the alignment with the ideal direction, (c) using gradients at the SAM point *decreases* the alignment with the noisy part.

## 4    WHY CAN SAM HELP AGAINST OVERFITTING OF NOISY LABELS?

In this section, we discuss another important aspect of SAM: its beneficial effect for the noisy label setting. We first analyze the development of the gradient direction over training and then interpret SAM as a *gradient reweighting schemes* where the gradients are reweighted according to the sharpness of the loss.

**Overfitting on noisy labels and the effect of SAM.** By the *noisy label setting* we refer to a situation where some fraction of the training labels is changed to uniformly random labels and kept fixed throughout the training.[1] This setting is challenging since standard training with ERM leads to *overfitting* as illustrated in Fig. 3a. In particular, it is commonly observed (Liu et al., 2020) that at the beginning of the training ERM fits the correctly labeled training points but, after some point, starts to also fit the incorrectly labeled points, which coincides with overfitting in terms of the test error. Thereby, the noisy label setting is different from the standard setting from Sec. 3.3: training to zero loss is harmful and leads to overfitting, and thus early stopping is needed (Li et al., 2020b). Fig. 3a illustrates that early stopping indeed leads to a significantly better test error for the ERM model compared to the model taken at the last epoch. At the same time, SAM noticeably improves generalization over ERM, although later in training SAM also starts to fit the noisy points and thus also requires early stopping, either explicitly via a (potentially noisy) validation set or implicitly via restricting the number of training epochs as in Foret et al. (2021).

**Benefits of following the gradient direction given by SAM.** Consider a batch of points $\mathcal{S}_{all} := \{(x, y)\}_{i=1}^{n}$ where some labels $y$ are noisy. We define $\mathcal{S}_{clean}$ and $\mathcal{S}_{noisy}$ to be the subsets of $\mathcal{S}_{all}$ with clean and noisy labels respectively. We also define $\mathcal{S}_{true}$ to be the same set of points as $\mathcal{S}_{clean}$ but where all the labels are correct. We denote by $L_{\mathcal{S}}(w) = \frac{1}{|\mathcal{S}|} \sum_{(x,y) \in \mathcal{S}} \ell_{x,y}(w)$ the loss on a set of points $\mathcal{S}$ where $w$ denotes the model parameters. The gradient update then can be decomposed as:

$$\nabla L_{\mathcal{S}_{all}}(w) = \frac{|\mathcal{S}_{clean}|}{|\mathcal{S}_{all}|} \nabla L_{\mathcal{S}_{clean}}(w) + \frac{|\mathcal{S}_{noisy}|}{|\mathcal{S}_{all}|} \nabla L_{\mathcal{S}_{noisy}}(w).$$

Here we want to analyze the contribution of each part of the gradient for the ERM and SAM weight updates. For this, in Fig. 3 (b) and (c), we compare the following directions over training epochs of an ERM model: $\nabla L_{\mathcal{S}_{all}}(w)$ vs. $\nabla L_{\mathcal{S}_{true}}(w)$ and $\nabla L_{\mathcal{S}_{all}}(w)$ vs. $\nabla L_{\mathcal{S}_{noisy}}(w)$ for the current point $w$ (i.e., the ERM update) and the point $w + \epsilon_{SAM}$ (i.e., the SAM update). To be robust to the label noise, we would like the update direction of an algorithm to be maximally aligned with the ideal direction $\nabla L_{\mathcal{S}_{true}}(w)$ and minimally aligned with the noisy direction $\nabla L_{\mathcal{S}_{noisy}}(w)$. The key observation of Fig. 3b and Fig. 3c is that SAM is changing the direction of the gradient by making it *less aligned with the noisy direction* at the later stage of training when the model starts to fit the label noise. Therefore, the SAM update is less dominated by the noisy labels and can resist overfitting longer. To better understand this effect, we discuss next the behavior of SAM when training a linear classifier in a binary classification setting.

**SAM improves generalization even for linear models.** Since using SumMax SAM (Eq. (2)) leads to better generalization than MaxSum SAM, we consider this technique for an overparametrized

---

[1]Thus, it is different from the setting where *new* label noise is added on each iteration (HaoChen et al., 2020).

linear model $f_x(w) = \langle w, x \rangle$ trained with the cross-entropy loss $\ell$ using gradient descent. The training objective of SumMax SAM then becomes:

$$\min_{w \in \mathbb{R}^d} \frac{1}{n} \sum_{i=1}^n \max_{\|\delta\|_2 \leq \rho} \ell\left(y_i \langle w + \delta, x_i \rangle\right) = \min_{w \in \mathbb{R}^d} \frac{1}{n} \sum_{i=1}^n \ell\left(y_i \langle w, x_i \rangle - \rho \|x_i\|_2\right). \quad (6)$$

We consider the following simple dataset: $x \sim \mathcal{N}(y\mu, \sigma^2 I_d)$, where $y \in \{-1, 1\}$ is the label, and $\mu$ is some unit vector. We use $\sigma = 0.1$, input dimension $d = 100$, $n = 50$ training samples, and we randomize 90% of the training labels. We select the best step sizes for ERM and SAM via a grid search. We can see from Fig. 4 that SAM with early stopping consistently (over ten random seeds) improves the test error over ERM. This illustrates that benefits of SAM extend beyond non-convex learning tasks with multiple minima and suggests that the linear setting can be a good starting point for understanding the benefit of SAM.

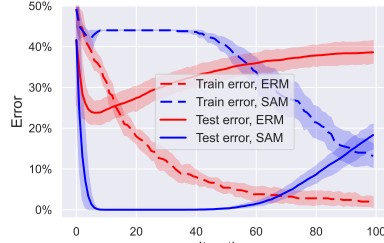

**Figure 4:** SAM for a linear model on two Gaussians with 90% label noise.

**SAM as a gradient reweighting scheme.** For a general non-linear binary classifier $f_x(w)$, on every iteration of gradient descent, the update direction is given by the gradient on all points:

$$\nabla L_{\mathcal{S}_{all}}(w) = \frac{1}{n} \sum_{i=1}^n \underbrace{\ell'(y_i \cdot f_{x_i}(w))}_{\text{loss derivative}} \cdot \underbrace{\nabla f_{x_i}(w)}_{\text{direction}}. \quad (7)$$

When we perturb the weights with SAM, we change in general both the loss derivative $\ell'(y \cdot f_x(w))$ and the direction given by the classifier's gradient $\nabla f_x(w)$. However, for a linear classifier $f_x(w)$, only $\ell'(y \cdot f_x(w))$ is changed while the direction $\nabla f_x(w) = x$ is independent of the weights. The fact that SAM helps even in this simple setting suggests that its crucial effect can lie only in modifying the loss derivative $\ell'(y \cdot f_x(w))$ which leads to beneficial *reweighting* of the per-example gradients of noisy and clean inputs. We note that gradient reweighting is also the idea behind robust losses (Ghosh et al., 2017; Zhang & Sabuncu, 2018) as they also affect only the loss derivative while keeping the direction $\nabla f_{x_i}(w)$ the same. This suggests the following interpretation of SumMax SAM:

> **Gradient reweighting interpretation of SAM**: SumMax SAM reweights per-example gradients amplifying the derivative of clean points *more* than the derivative of noisy points so that the overall gradient direction $\nabla L_{\mathcal{S}_{all}}(w)$ is more aligned with $\nabla L_{\mathcal{S}_{clean}}(w)$.

We observe from Eq. (6) that SAM offsets the margin $y_i \langle w, x_i \rangle$ of each example by $\rho \|x_i\|_2$ More generally, using SumMax SAM for non-linear models guarantees that the offset is *non-negative* for each example as maximization of a monotonic loss $\ell$ implies minimization of the margin. Note that this would not be the case for MaxSum SAM or average-case perturbations of the weights. We discuss next why offsetting the loss can be valuable to reweight the gradients of different examples.

We first note that the *early-learning phenomenon* which is proven to hold under some assumptions in Liu et al. (2020) implies that the correctly labeled points are fitted first during training so that at some early stage they tend to have a higher margin than the noisy points. We illustrate this in Fig. 5 for a linear model. We then consider the following two losses: cross-entropy loss (CE) and generalized cross-entropy loss (GCE) (Zhang & Sabuncu, 2018) (we hide the dependency of $f$ on $w$ for brevity)

$$\ell_{CE}(y \cdot f_x) = \log(1 + \exp(-y \cdot f_x)), \qquad \ell_{GCE}(y \cdot f_x) = 1/q \left(1 - (1 + \exp(-y \cdot f_x))^{-q}\right).$$

We plot these losses in Fig. 6a (using $q = 0.7$ for GCE following Zhang & Sabuncu (2018)) together with the derivative change from adding an offset $c$ in Fig. 6b and 6c. We observe based on Fig. 6b that the offset affects more significantly the points with small positive margins (i.e., mostly correctly labeled points due to the early-learning phenomenon) whose derivative is amplified more making it closer to its minimal value of -1. Fig. 6c suggests that GCE has a better effect than CE since GCE does not only decrease the derivative of points with small positive margin but it also increases the derivative of the points with small negative margin (i.e., mostly noisy points) making it closer to zero. This suggests that combining GCE with SAM can lead to further benefits which is indeed confirmed experimentally in App. D.

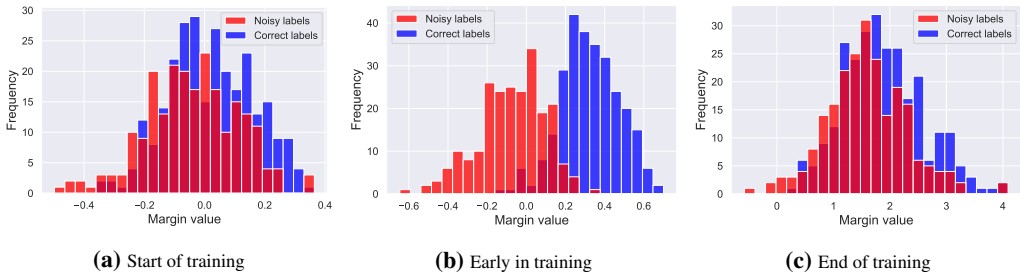

**(a)** Start of training  **(b)** Early in training  **(c)** End of training

**Figure 5:** An illustration of the early-learning phenomenon (Liu et al., 2020) on the training set for a linear model. Early in training the model achieves a good separation between noisy and correctly labeled points which disappears towards the end of training.

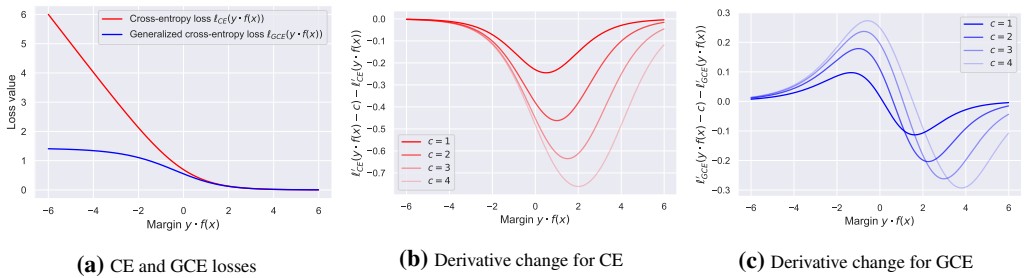

**(a)** CE and GCE losses  **(b)** Derivative change for CE  **(c)** Derivative change for GCE

**Figure 6:** Introducing a loss offset $c$ mainly affects points with small positive (for CE and GCE) and small negative margin (for GCE) which mostly correspond to clean and noisy points, respectively, due to the early-learning phenomenon.

**Table 1:** An ablation study for different variations of SAM on a two-class CIFAR-10 dataset. The error rates are averaged over three random seeds and reported with the standard deviation

| NAME | GRADIENT FORMULA $L_{\mathcal{S}_{all}}(w)$ | ERROR RATE |
|---|---|---|
| ERM | $\frac{1}{n}\sum_{i=1}^{n}\ell'(y_i \cdot f_{x_i}(w))\nabla f_{x_i}(w)$ | $13.47\% \pm 0.80$ |
| SAM DIRECTION | $\frac{1}{n}\sum_{i=1}^{n}\ell'(y_i \cdot f_{x_i}(w))\nabla f_{x_i}(w_{SAM})$ | $11.30\% \pm 0.68$ |
| SAM DERIVATIVE | $\frac{1}{n}\sum_{i=1}^{n}\ell'(y_i \cdot f_{x_i}(w_{SAM}))\nabla f_{x_i}(w)$ | $7.45\% \pm 0.28$ |
| SAM | $\frac{1}{n}\sum_{i=1}^{n}\ell'(y_i \cdot f_{x_i}(w_{SAM}))\nabla f_{x_i}(w_{SAM})$ | $9.02\% \pm 0.34$ |

**Does the loss derivative change explain the behavior of a non-linear model?** To further support our gradient reweighting interpretation for non-linear models, we show that changing only the loss derivative $\ell'(y_i \cdot f_{x_i}(w_{SAM}))$ (see Eq. (7)) and discarding the change of the direction leads to the main benefit in SAM. For this, we train a ResNet-18 on a two-class CIFAR-10 under 80% label noise with early stopping and study different ways to modify the gradient $\nabla L_{\mathcal{S}}(w)$ used on each iteration of training. We report results in Table 1 where we observe that SAM DIRECTION leads only to a small improvement compared to ERM ($13.47\%$ to $11.30\%$) while SAM DERIVATIVE is helping most leading to $7.45\%$ test error. This gives more evidence that the main improvement from using SAM comes from changing the derivative of the loss $\ell'(y_i \cdot f_{x_i}(w_{SAM}))$ and not the direction $\nabla f_{x_i}(w_{SAM})$ which is coherent with our gradient reweighting interpretation of SAM. We further report extra experimental details and the plots over epochs in App. D.

## 5 WHY CAN SAM HELP AGAINST ROBUST OVERFITTING?

Here we draw parallels between the overfitting observed in the noisy label setting and in robust training (Rice et al., 2020) where SAM also improves generalization (Wu et al., 2020).

**The effect of SAM is similar for robust overfitting.** In Fig. 7, we plot the error under adversarial perturbations obtained via the PGD attack (we use 10 iterations, see details in App. D) for an $\ell_\infty$ adversarially trained model (Madry et al., 2018) with $\epsilon = 8/255$ trained with ERM and SAM for 1000

epochs. For this experiment, we use the SAM algorithm as in Foret et al. (2021) which also leads to similar improvements of robust error as the method introduced in Wu et al. (2020) which relies on a different scaling of the layerwise perturbation bounds. We observe that, when trained for sufficiently long (contrary to Wu et al. (2020) that only train for 200 epochs), *SAM also leads to overfitting*, showing high similarity to the noisy label setting (Fig. 3a).

**Implicit label noise in adversarial training.** Related to the point above, a recent work of Dong et al. (2021) argues about the presence of inputs which are "noisy" for adversarial training, and whose removal mitigates robust overfitting. They further discuss that such inputs become *ambiguous* for a human observer after adding $\ell_\infty$ adversarial perturbations. The problem stems from the formulation of adversarial examples which is widely used: $\max_{\|\delta\|_\infty \le \epsilon} \ell(x + \delta)$ instead of the *label-preserving* formulation $\max_{\|\delta\|_\infty \le \epsilon,\, y(x)=y(x+\delta)} \ell(x + \delta)$ which is obviously impractical since we cannot query the true label $y(x + \delta)$ at an arbitrary point during training. Thus, the commonly used adversarial perturbations can change the effective label $y(x + \delta)$ leading to *implicit label noise*. This can be confirmed by implementing label-preserving adversarial training for a linear model where we know the ground truth model. We can use the same two Gaussian example as in Fig. 4 but without explicit label noise and perform adversarial training in the subspace orthogonal to the discriminative direction $\mu$ which ensures that the label of $x + \delta$ is not affected. We show experimentally in Fig. 8 that (1) robust overfitting can be observed also in simple linear models and (2) using the label-preserving formulation of adversarial training prevents it. This suggests that robust overfitting is strongly connected to the labels of adversarial examples.

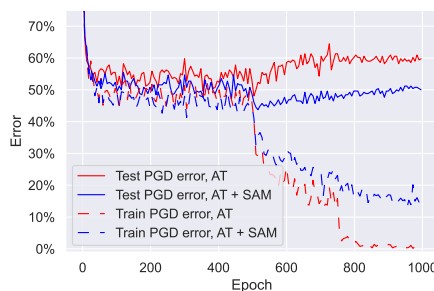

**Figure 7:** The effect of SAM on robust overfitting in adversarial training is similar to its effect for noisy labels (cf. Fig. 3a).

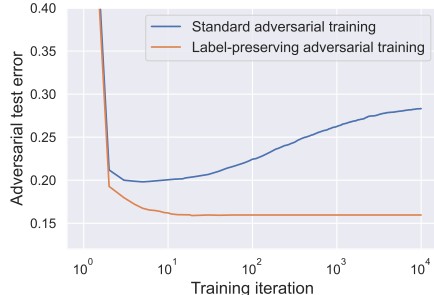

**Figure 8:** Robust overfitting for a linear model is not observed for label-preserving adversarial training.

**Mitigating implicit label noise by SAM and other methods.** Performing label-preserving adversarial training requires the knowledge of the ground truth labeling function which is impractical. Thus, in practice one has to rely on algorithms that prevent fitting the implicit label noise in other ways. We hypothesize that an early-learning phenomenon similar to the one observed in training with noisy labels (Liu et al., 2020) should also hold for adversarial training, and our gradient reweighting explanation of SAM from Sec. 4 should also extend to this setting. We also note that other solutions against robust overfitting that have been recently proposed in the literature—such as early stopping (Rice et al., 2020), weight averaging (Gowal et al., 2020; Rebuffi et al., 2021), bootstrap (Chen et al., 2021), data removal (Dong et al., 2021)—also rely on the early-learning phenomenon and have been previously explored in the noisy label literature. We confirm this further on GCE (Zhang & Sabuncu, 2018) and semi-supervised pairing terms (Luo et al., 2019) in App. D. This further suggests that both settings are closely related, and improvements in one setting can also translate to the other one which can explain the improvement from SAM against robust overfitting.

## 6 CONCLUSIONS AND OUTLOOK

We have discussed two different objectives motivated by SAM and analyzed their implicit bias for a diagonal linear network. Then we have provided a convergence proof for the variant of SAM used in practice and confirmed empirically that the best generalization performance is achieved at a solution with nearly zero training loss on a standard image classification tasks. At the same time, in the noisy label setting, it is harmful to achieve low training loss even with SAM since this would involve fitting the noisy labels. We have discussed why SAM can be helpful in this case: we observed that SAM amplifies the contribution of the correctly labeled points compared to the noisy ones in the gradient update. Finally, we argue that the overfitting observed in noisy label setting shares a lot of similarities with overfitting observed in adversarial training which can explain why SAM is also helpful there.

## REPRODUCIBILITY STATEMENT

**Theoretical results.** We provide proofs of our theoretical results in the Appendix in Sec. A for the implicit bias and in Sec. B for the convergence of different variants of SAM. The assumptions **A1**-**A3** for the convergence proofs are mentioned at the beginning of Sec. 3.2. An additional assumption **A2'** is used only for the statements in the appendix and thus is introduced only there.

**Empirical results.** To facilitate reproducibility of our experiments, in Sec. C we specify all the hyperparameters of our experiments on deep networks and linear models. Moreover, in Sec. D, we provide more details about some of the experiments from the main part.

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

# Appendix

ORGANIZATION OF THE APPENDIX

The appendix contains proofs related to sharpness and convergence of different variants of SAM that we omitted from the main part of the paper due to the space constraints. Moreover, the appendix also contains additional implementation details and extended experimental results. The appendix is organized as follows:

- Sec. A: proofs related to the implicit bias of SumMax and MaxSum SAM.
- Sec. B: proofs related to the convergence of different variants of SAM.
- Sec. C: experimental details for the epxeriments with deep networks and linear models.
- Sec. D: additional experiments about the importance of the change in the loss derivative for deep networks, improvement of standard generalization under label noise by adversarial training, and the experiment with the GCE loss and semi-supervised pairing term that mitigate robust overfitting.

## A    THEORETICAL ANALYSIS OF THE IMPLICIT BIAS

To understand why SAM is generalizing better than ERM we consider the simpler problem of noiseless regression with 2-layer diagonal linear network for which we can precisely characterize the implicit bias of the optimizations algorithms in hand.

**Overparametrised noiseless regression.**    We consider linear regression with inputs, outputs $(x_1, y_i, \ldots, x_n, y_n) \in (\mathbb{R}^{d+1})^n$. We consider the overparametrised case and assume that there exists at least one interpolating parameter $\beta^* \in \mathbb{R}^d$ which fits the training set: $y_i = \langle \beta^*, x_i \rangle$ for all $1 \leq i \leq n$. We study the quadratic loss for which the training loss is written as:

$$L(\beta) := \frac{1}{4n} \sum_{i=1}^n (\langle \beta, x_i \rangle - y_i)^2 = \frac{1}{4n} \sum_{i=1}^n \langle \beta - \beta^*, x_i \rangle^2.$$

2**-layer diagonal linear network.**    The simplest parametrisation of $\beta$ is to consider $\beta = w$ which corresponds to the classical least-squares framework. In this case, first order methods (GD, with and without momentum) converge towards the same solution: they have the same implicit bias. The same holds for the variants of SAM we consider and the linear case is not helping us to distinguish them and to show their primacy on ERM. To theoretically resolve this question, we study a simple non-linear parametrisation: the quadratic parametrisation $\beta = w_+^2 - w_-^2$ with $w = [w_+, w_-]^\top \in \mathbb{R}^{2d}$. The training loss in $w$ is therefore written:

$$L(w) := \frac{1}{4n} \sum_{i=1}^n (\langle w_+^2 - w_-^2, x_i \rangle - y_i)^2 \frac{1}{4n} \sum_{i=1}^n (\langle \beta, x_i \rangle - y_i)^2 \ \text{ with } \ \beta = w_+^2 - w_-^2. \quad (8)$$

This parametrisation can be viewed as a simplified linear network of depth 2. Additionally to being a toy neural model, this model has received recently a lot of attention since it is possible to precisely characterize the implicit bias of descent algorithm on it.

**Optimization algorithms.**    We consider minimizing the loss $L(w)$ using three different algorithms:

- Gradient descent with infinitesimally small stepsize (in the gradient flow limit):
$$\dot{w}_t = -\nabla L(w_t). \quad (9)$$
- The MaxSum SAM algorithm with infinitesimally small outer-stepsize and inner step size $\rho \geq 0$:
$$\dot{w}_t = -\nabla L(w_t + \rho \nabla L(w_t)). \quad (10)$$
- The SumMax SAM with infinitesimally small outer-stepsize and inner step size $\rho \geq 0$:
$$\dot{w}_t = -\frac{1}{n} \sum_{i=1}^n \nabla \ell_i(w_t + \rho \nabla \ell_i(w_t)) \quad (11)$$

**Implicit bias of the Gradient flow.**    Let us define the function $\phi_\alpha$ for $\alpha \in \mathbb{R}^d$ as

$$\phi_\alpha(\beta) = \sum_{i=1}^d \alpha_i^2 q(\beta_i/\alpha_i^2) \text{ where } q(z) = \int_0^z \text{arcsinh}(u/2)du = 2 - \sqrt{4 + z^2} + z \, \text{arcsinh}(z/2).$$
(12)

Note that if we have $\alpha \in \mathbb{R}$ then we consider $\phi_\alpha = \phi_{\alpha 1}$. We will make great uses of this function to precisely characterize the implicit bias of the different algorithms we consider. Indeed, following Woodworth et al. (2020) we can show the following result for the gradient flow dynamics (9)

**Proposition 3** (Theorem 1 of Woodworth et al. (2020))**.** *For any $0 < \alpha < \infty$, if the solution $\beta_\infty$ of the gradient flow (9) started from $w_0 = \alpha 1$ for the squared parameter problem in Eq.(8) satisfies $X\beta_\infty = y$, then*

$$\beta_\infty = \arg \min_{\beta \in \mathbb{R}^d} \phi_\alpha(\beta) \ \ s.t. \ \ X\beta = y,$$
(13)

*where $\phi_\alpha$ is defined in Eq.(12).*

It worth noting that the implicit regularizer $\phi_\alpha$ interpolates between the $\ell_1$ and $\ell_2$ norms (see Woodworth et al., 2020, Theorem 2). Therefore an implicit bias depending on the scale of the initialization is observed. The algorithm, started from $\alpha 1$, converges to the minimum $\ell_1$-norm interpolator for $\alpha$ small and to the minimum $\ell_2$-norm interpolator for $\alpha$ large. The proof follows from (a) the KKT condition for the optimization problem (13): $\nabla \phi_\alpha(w) = X^\top \nu$ for a Lagrange multiplier $\nu$ and (b) the closed form solution obtained by integrating the gradient flow, $w = b(X^\top \nu)$ for some function $b$ and some vector $\nu$. Identifying $\nabla \phi_\alpha(w) = b^{-1}(w)$ leads to the solution.

Considering the same proof technique, we now derive the implicit bias of both variants of SAM algorithms.

**Implicit bias of MaxSum SAM algorithm.**    We characterize the implicit bias of the MaxSum dynamics (10) in the following proposition. Recall that the function $\phi_\alpha$ is defined in Eq. (12).

**Proposition 4.** *For any $0 < \alpha < \infty$, if the solution $\beta_\infty$ of the gradient flow (10) started from $w_0 = \alpha 1$ for the squared parameter problem in Eq.(8) satisfies $X\beta_\infty = y$, then*

$$\beta_\infty = \arg \min_\beta \phi_{\alpha_{MaxSum}}(\beta) \ \ s.t. \ \ X\beta = y,$$

*where $\alpha_{MaxSum} = \alpha e^{-\frac{\rho}{4n^2} \int_0^\infty (X^\top r_s)^2 ds + O(\rho^2)}$.*

We note that for $\rho$ small enough $\alpha_{\text{MaxSum}} < \alpha$. The scale of the vector $\frac{1}{n^2} \int_0^\infty (X^\top r_s)^2 ds$ (which influences the bias effect) is related to the loss integral $\frac{d}{n} \int_0^\infty L(w(s))ds$ since $\|r_s\|^2 = nL(w(s))$ (see intuition in Eq.(16)). Thereby the speed of convergence of the loss controls the magnitude of the biasing effect. However in the case of MaxSum SAM, as explained below, this effect is typically negligible because of the extra prefactor $\frac{d}{n}$ and this implementation behaves similarly as ERM (as shown in the experiments in Sec. 3.1).

*Proof.* We follow the proof technique of Woodworth et al. (2020). We first derive the equation satisfies by the flow

$$\dot{w}(t) = -\frac{1}{2n} \tilde{X}^\top r_+(t) \circ w_+(t)$$
$$= -\frac{1}{2n} \tilde{X}^\top r_+(t) \circ (w(t) + \frac{\rho}{2n} \tilde{X}^\top r(t) \circ w(t)),$$

where $w_+(t) = w(t) + \rho \nabla L(w(t)$ is the intermediate SAM step, $r(t) = \tilde{X}w(t)^2 - y$ and $r_+(t) = \tilde{X}w_+(t)^2 - y$ are the residual of $w(t)$ and $w_+(t)$ and we denote by $\tilde{X} = [X \ -X]$. We can directly integrate this ODE to obtain a closed form expression for $w(t)$:

$$w(t) = w(0) \circ \exp(-\frac{1}{2n} \tilde{X}^\top \int_0^t r_+(s)ds) \circ \exp(-\frac{\rho}{4n^2} \int_0^t (\tilde{X}^\top r_+(s)) \circ (\tilde{X}^\top r(s))ds).$$

Using that the flow is initialized at $w(0) = \alpha 1$ and the definition of $\beta(t)$ yields to

$$\beta(t) = u(t)^2 - v(t)^2$$

$$= \alpha^2 \exp(-\frac{1}{n} X^\top \int_0^t r_+(s)ds) \circ \exp(-\frac{\rho}{2n^2} \int_0^t (X^\top r_+(s)) \circ (X^\top r(s))ds)$$

$$- \alpha^2 \exp(\frac{1}{n} X^\top \int_0^t r_+(s)ds) \circ \exp(-\frac{\rho}{2n^2} \int_0^t (X^\top r_+(s)) \circ (X^\top r(s))ds)$$

$$= 2\alpha^2 \exp(-\frac{\rho}{2n^2} \int_0^t (X^\top r_+(s)) \circ (X^\top r(s))ds) \sinh(-\frac{1}{n} X^\top \int_0^t r_+(s)ds).$$

Recall we are assuming that $\beta_\infty$ is global minimum of the loss, i.e., $X\beta_\infty = y$. Thus

$$X\beta_\infty = y$$

$$\beta_\infty = b_{\alpha_{\mathrm{MaxSum}}}(X^\top \nu),$$

with and $b_\alpha(z) = 2\alpha^2 \circ \sinh(z)$, $\nu = -\frac{1}{n}\int_0^\infty r_+(s)ds$ and

$$\alpha_{\mathrm{MaxSum}} = \alpha \exp(-\frac{\rho}{4n^2} \int_0^\infty (X^\top r_+(s)) \circ (X^\top r(s))ds). \tag{14}$$

Denoting by

$$\nabla \phi_\alpha(\beta) = b_\alpha^{-1} \operatorname{arcsinh}(\frac{1}{2\alpha^2} \circ \beta),$$

and integrating this equation, we obtain the expression of $\phi_\alpha = \sum_{i=1}^d \alpha_i^2 q(\beta_i/\alpha_i^2)$ where $q(z) = \int_0^z \operatorname{arcsinh}(u/2)du = 2 - \sqrt{4 + z^2} + z\operatorname{arcsinh}(z/2)$.

Therefore $\beta_\infty$ satisfies the KKT conditions for the minimum norm interpolator problem $\arg\min_\beta \phi_\alpha(\beta)$ s.t. $X\beta = y$, i.e.

$$X\beta_\infty = y$$

$$\nabla \phi_\alpha(\beta_\infty) = X^\top \nu.$$

This proves the first part of the result. Now using the definition of $r_+(s)$ we obtain

$$r_+(t) = \tilde{X}w_+(t)^2 - y$$

$$= \tilde{X}(w(t) + \rho\tilde{X}^\top r(t) \circ w(t))^2 - y$$

$$= r(t) + 2\rho\tilde{X}(\tilde{X}^\top r(t)) \circ w(t)^2 + \rho^2\tilde{X}(\tilde{X}^\top r(t))^2 \circ w(t)^2$$

$$= r(t) + 2\rho X(X^\top r(t)) \circ (u(t)^2 + v(t)^2) + \rho^2 X(X^\top r(t))^2 \circ \beta(t).$$

And

$$X^\top r_+(t) = X^\top r(t) + O(\rho),$$

which concludes the second part of the result. □

**Implicit bias of SumMax SAM algorithm** We characterize similarly the implicit bias of the SumMax dynamics (11) in the following proposition. Recall that the function $\phi_\alpha$ is defined in Eq. (12).

**Proposition 5.** *For any $0 < \alpha < \infty$, if the solution $\beta_\infty$ of the gradient flow (11) started from $w_0 = \alpha 1$ for the squared parameter problem in Eq.(8) satisfies $X\beta_\infty = y$, then*

$$\beta_\infty = \arg\min_\beta \phi_{\alpha_{SumMax}}(\beta) \quad s.t. \ X\beta = y,$$

*where $\alpha_{SumMax} = \alpha e^{-\frac{\rho}{4n}\int_0^\infty \sum_{i=1}^n x_i^2(x_i^\top \beta(s) - y_i)^2 ds + O(\rho^2)}$.*

*In addition, if we assume that there exist $R, B \geq 0$ such that the features are bounded $\|x\| \leq R$ almost surely and that the trajectory of the flow is bounded $\|\beta(t)\| \leq B$ for all $t \geq 0$, we have for all $\rho \leq \frac{1}{R^2\sqrt{B(B+\|\beta_*\|)}}$:*

$$\alpha_{SumMax} \leq \alpha$$

We note that for $\rho$ smaller than an explicit constant depending on the problem quantities, we can assert that $\alpha_{\text{MaxSum}} < \alpha$. Furthermore the scale of the vector $\frac{1}{n}\int_0^\infty \sum_{i=1}^n x_i^2(x_i^\top\beta(s) - y_i)^2 ds$ (which influences the bias effect) is related to the loss integral $d\int_0^\infty L(w(s))ds$, through $\|r_s\|^2 = nL(w(s))$ (see intuition Eq. (17)). Thus the speed of convergence of the loss controls the magnitude of the biasing effect. This effect is typically important in the case of SumMax SAM (see experiments Sec.3.1).

*Proof.* The proof follows the same lines as the proof of Proposition 4 Let us denote by $\tilde{x}_i = [x_i - x_i]$. We then have that the dynamics of the flow (11) satisfies

$$\dot{w}(t) = -\frac{1}{2n}\sum_{i=1}^n \tilde{x}_i r_{i,+}(t) \circ w_+(t)$$

$$= -\frac{1}{2n}\sum_{i=1}^n \tilde{x}_i r_{i,+}(t)(1 + \frac{\rho}{2}\tilde{x}_i r_i(t)) \circ w(t),$$

where $w_+(t)$ is the intermediate SAM step, $r_i(t) = \tilde{x}_i^\top w(t)^2 - y_i$ and $r_{i,+}(t) = \tilde{x}_i w_+(t)^2 - y_i$ are the residual of $w(t)$ and $w_+(t)$ at the observation $(x_i, y_i)$. Integrating we obtain:

$$w(t) = w(0)\circ\exp(-\frac{1}{n}\tilde{X}^\top\int_0^t r_+(s)ds)\circ\exp(-\frac{\rho}{2n}\sum_{i=1}^n\tilde{x}_i^2\int_0^t r_{i,+}(s)r_i(s)ds).$$

The remainder of the proof is similar to the one of Proposition 4 and we directly obtain that

$$\alpha_{\text{SumMax}} = \alpha\exp(-\frac{\rho}{4n}\sum_{i=1}^n\tilde{x}_i^2\int_0^t r_{i,+}(s)r_i(s)ds). \tag{15}$$

Using the definition of $r_{i,+}(t)$ we have

$$r_{i,+}(t) = \tilde{x}_i^\top w_+(t)^2 - y_i$$
$$= \tilde{x}_i^\top w(t)^2 \circ (1 + \rho r_{i,t}\tilde{x}_i)^2 - y_i$$
$$= \tilde{x}_i^\top w(t)^2 \circ (1 + 2\rho r_{i,t}\top x_i + \rho r_{i,t}^2\top x_i^2) - y_i$$
$$= r_{i,t} + 2\rho r_{i,t}(u(t)^2 + v(t)^2)^\top x_i^2 + \rho^2 r_{i,t}^2\beta(t)^\top x_i^3.$$

And therefore

$$x_i^2 r_{i,+}(t)r_i(t) = x_i^2 r_i(t)^2[1 + 2\rho(u(t)^2 + v(t)^2)^\top x_i^2 + \rho^2 r_i(t)\beta(t)^\top x_i^3])$$

So if $\rho$ is such that $1 + \rho^2 r_i(t)\beta(t)^\top x_i^3 \geq 0$ then $\alpha_{\text{SumMax}} < \alpha$. Using Cauchy-Schwarz inequality $|r_i(t)\beta(t)^\top x_i^3| \leq \|x_i\|(\|\beta\| + \|\beta_*\|)\|\beta\|\|x_i^3\|$ and Holder inequality $\|x_i^3\| \leq \|x_i\|^3$. Thus $|r_i(t)\beta(t)^\top x_i^3| \leq \|x_i\|^4(\|\beta\| + \|\beta_*\|)\|\beta\|$. And we have the result for $\rho \leq R^{-2}(B(B + \|\beta_*\|))^{-1/2}$. $\qquad\square$

**Comparison between SumMax and MaxSum implicit bias and connection with the loss integral** We wish to compare the two leading terms $I_{\text{MaxSum}}(t) = \frac{1}{n^2}(X^\top r(t))^2 = \frac{1}{n^2}(\sum_{i=1}^n x_i r_i)^2$ and $I_{\text{SumMax}}(t) = \frac{1}{n}\sum_{i=1}^n x_i^2 r_i(t)^2$ which comprise the first order in the expansion of the $\alpha$ and relate them to the loss value at $w(t)$.

We first note using Cauchy-Schwarz inequality that $I_{\text{SumMax}}(t) \geq I_{\text{MaxSum}}(t)$ but we aim at obtaining a more quantitative result, even though the derivation will be informal. Comparing their $\ell_1$-norms amounts to compare the empirical fourth moment $\|I_{\text{MaxSum}}(t)\|_1 = (w - w_*)^\top\frac{1}{n}\sum_{i=1}^n\|x_i\|^2 x_i x_i^\top(w - w_*)$ and second moment squared $\|I_{\text{SumMax}}(t)\|_1 = (w - w_*)^\top[\frac{1}{n}\sum_{i=1}^n x_i x_i^\top]^2(w - w_*)$.

We can compare the typical operator norm of these two random matrices. Following the Bai-Yin's law, the operator norm of a Wishart matrix is with high probability $\|\frac{1}{n}\sum_{i=1}^n x_i x_i^\top\| \approx \frac{d}{n}$ and that

with high-probability, the squared norm of a Gaussian vector is $\|x_i\| \approx d$. Therefore we obtain that

$$\|\frac{1}{n}\sum_{i=1}^n \|x_i\|^2 x_i x_i^\top\| \approx d\|\frac{1}{n}\sum_{i=1}^n x_i x_i^\top\| \approx \frac{d^2}{n}$$

$$\|[\frac{1}{n}\sum_{i=1}^n x_i x_i^\top]^2\| \leq \|[\frac{1}{n}\sum_{i=1}^n x_i x_i^\top]\|^2 \approx \left(\frac{d}{n}\right)^2$$

Therefore in the overparametrized regime ($d >>> n$) we typically have that $\frac{I_{\text{SumMax}}(t)}{I_{\text{MaxSum}}(t)} \approx n$ and the biasing effect of the SumMax implementation would tend to be O(n) time better than the biasing effect of the MaxSum implementation.

However this first insight only enables to compare $I_{\text{MaxSum}}(t)$ and $I_{\text{SumMax}}(t)$. It is not informative on the intrinsic biasing effect of MaxSum and SumMax SAM. With this aim, we would like to relate the quantities $I_{\text{MaxSum}}(t)$ and $I_{\text{SumMax}}(t)$ to the loss function evaluated in $w(t)$. With high probability (still using the concentration of a Gaussian vector) we have that

$$\|I_{\text{MaxSum}}(t)\|_1 = (w(t) - w_*)^\top \frac{1}{n}\sum_{i=1}^n \|x_i\|^2 x_i x_i^\top (w(t) - w_*)$$

$$\approx d(w(t) - w_*)^\top \frac{1}{n}\sum_{i=1}^n x_i x_i^\top (w(t) - w_*)$$

$$= dL(w(t)). \tag{16}$$

And using the concentration of Wishart matrices, i.e., $\frac{1}{d}[XX^\top] \approx I$ for large dimension $d$, we also have

$$\|I_{\text{SumMax}}(t)\|_1 = \frac{1}{n^2}(w(t) - w_*)^\top X^\top X X^\top X(w(t) - w_*)$$

$$\frac{d}{n^2}(w(t) - w_*)^\top X^\top \frac{1}{d}[XX^\top] X(w(t) - w_*)$$

$$\approx \frac{d}{n}(w(t) - w_*)^\top \frac{1}{n}[X^\top X](w(t) - w_*)$$

$$= \frac{d}{n}L(w(t)). \tag{17}$$

These approximations provide some intuitions on why the biasing effect of both SAM implementation can be related to the integral of the loss. We let a formal derivation of these results as future work.

**Additional experiments on implicit bias of stochastic methods.** We provide an additional experiment to investigate the performance of stochastic implementation of the ERM, MaxSum and SumMax SAM. As explained by Pesme et al. (2021), we observe in Fig.9 that the stochastic implementations enjoy a better implicit bias than their deterministic counterparts. We note that the fact that small batch versions generalize better than full batch version is commonly observed in practice for deep networks Keskar et al. (2016). We let the characterization of the implicit bias of these stochastic implementations as future works.

# B    CONVERGENCE OF SAM ALGORITHM

In this section we prove the convergence of the SAM algorithm to a stationary point of the function $L$ or to a minimum when the function is convex.

## B.1    CONVERGENCE OF DETERMINISTIC SAM ALGORITHM

Here we aim to understand if the one-step approximation of SAM makes sense from an *optimization perspective*, i.e., whether the algorithm proposed in (Foret et al., 2021) can successfully minimize the

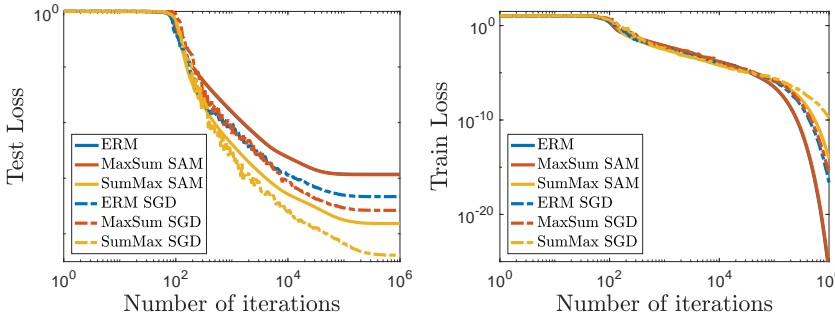

**Figure 9:** Implicit bias of SAM on a sparse regression problem with $d = 30$, $n = 20$, $x_i \sim \mathcal{N}(0, I)$, $\kappa = \|\beta_*\|_0 = 3$, $y_i = x_i^\top \beta_*$ and $f_{u,v}(x) = x^\top(u \odot v)$. All methods are initialized at $\alpha = 0.01$ and used with step-size $\gamma = 1/L$ and $\rho = 1/L$. SumMax SGD converges to a solution which generalizes better (left plot) and enjoys a different implicit bias from the other methods. At the same time, all algorithms converge to a global minimum of $f$ at linear rate (right plot). The convergence rate is inversely proportional to the biasing effect.

training objective. This is an important aspect since if an optimization algorithm cannot sufficiently minimize the training loss, the resulting classifier typically will not lead to good *generalization* performance, even if it has some useful implicit bias (Neyshabur et al., 2014). We first consider the following first-order optimization algorithm on the objective $L$ parameterized by $w$:

$$w_{t+1} = w_t - \gamma_t \nabla L(w_t + \rho_t \nabla L(w_t)), \tag{18}$$

where $\rho_t$ and $\gamma_t$ are the step sizes of the inner and outer steps, respectively. This update rule is reminiscent of the extra-gradient algorithm (Korpelevich, 1977) but with an *ascent* in the inner step, i.e., the gradient is evaluated not at the current iterate but at the point which maximises locally the function. Moreover, this update rule can also be seen as a realization of the general extrapolated gradient descent framework suggested in Lin et al. (2020). However, taking an *ascent* step for extrapolation is not discussed there, and the convergence properties of the update rule from equation 18, to the best of our knowledge, have not been proven.

**Summary of the convergence results.**     Let us first recall the definition of smoothness.

**(A2')** ($\beta$-smoothness). *It exists $\beta > 0$ such that $\|\nabla L(w) - \nabla L(v)\| \leq \beta \|w - v\|$ for all $w, v \in \mathbb{R}^d$.*

When the function $L$ is $\beta$-smooth, convergence to stationary points can be obtained.

**Proposition 6.** *Assume (A2'). For any $\gamma < 1/\beta$ and $\rho < 1/\beta$, the iterates equation 18 satisfy for all $T \geq 0$:*

$$\frac{1}{T} \sum_{t=0}^{T-1} \|\nabla L(w_t)\|^2 \leq \frac{2}{\gamma(1 - \rho\beta)T}(L(w_0) - L_*),$$

*If, in addition, the function $L$ satisfies (A3), then:*

$$L(w_T) - L_* \leq \left(1 - \frac{\gamma(1 - \rho\beta)\mu}{2}\right)^T (L(w_0) - L_*).$$

We can make the following remarks:

- We recover the rates of gradient descent but with constants increasing with the ascent step-size $\rho$.
- The condition $\rho < 1/\beta$ is necessary since the point $w + 1/\beta\nabla L(w)$ can be a local maximum of $L$. Such $w$ would be a fixed point of the algorithm without being a stationary point of $L$.
- The proof crucially relies on the bound $\langle \nabla L(w_t + \rho\nabla L(w_t)), \nabla L(w_t)\rangle \geq (1 - \rho\beta)\|\nabla L(w_t)\|^2$ which shows that SAM-step is well aligned with the gradient step (see Lemma 7) and on a descent inequality similar to the classical one for gradient descent (see Lemma 8).

- For non-convex functions, full details are provided in Proposition 9. When the function satifies an addition Polyak-Lojasiewicz inequality, details are in Proposition 10.
- For convex functions, $\langle \nabla L(w_t + \rho \nabla L(w_t)), \nabla L(w_t) \rangle \geq \|\nabla L(w_t)\|^2$ and convergence holds for any step-size $\rho$ given that $\gamma \rho$ is small enough. Details are provided in Proposition 11 .

**Auxillary Lemmas.** The following lemma shows that the SAM update is well correlated with the gradient $\nabla L(w)$ and will be cornerstone to our proof.

**Lemma 7.** *Let $L$ be a differentiable function and $w \in \mathbb{R}^d$. We have the following bound for any $\rho \geq 0$:*

$$\langle \nabla L(w + \rho \nabla L(w)), \nabla L(w) \rangle \geq (1 + \alpha \rho) \|\nabla L(w)\|^2 \ where \ \alpha = \begin{cases} -\beta & \text{if } L \text{ is } \beta\text{-smooth,} \\ 0 & \text{if } L \text{ is convex} \\ \mu & \text{if } L \text{ is } \mu\text{-strongly convex.} \end{cases}$$

*Proof.* We simply add and subtract a term $\|\nabla L(w)\|^2$ in order to make use of classical inequalities bounding $\langle \nabla L(w_1) - \nabla L(w_2), w_1 - w_2 \rangle$ by $\|w_1 - w_2\|^2$ for smooth or convex functions and $w_1, w_2 \in \mathbb{R}^d$.

$$\begin{aligned} \langle \nabla L(w + \rho \nabla L(w)), \nabla L(w) \rangle &= \langle \nabla L(w + \rho \nabla L(w)) - \nabla L(w), \nabla L(w) \rangle + \|\nabla L(w)\|^2 \\ &= 1/\rho \langle \nabla L(w + \rho \nabla L(w)) - \nabla L(w), \rho \nabla L(w) \rangle + \|\nabla L(w)\|^2 \\ &\geq (1 + \alpha \rho) \|\nabla L(w)\|^2, \end{aligned}$$

where the last inequality is using that

$$\langle \nabla L(w_1) - \nabla L(w_2), w_1 - w_2 \rangle \geq \alpha \|w_2 - w_1\|^2, \text{ where } \alpha = \begin{cases} -\beta & \text{if } L \text{ is } \beta\text{-smooth,} \\ 0 & \text{if } L \text{ is convex} \\ \mu & \text{if } L \text{ is } \mu\text{-strongly convex.} \end{cases}$$

$\square$

The next lemma shows that the decrease of function values of the SAM algorithm defined in Eq.(18) can be controlled similarly as in the case of gradient descent Nesterov (2004).

**Lemma 8.** *Assume (A2'). For any $\gamma \leq 1/\beta$, the iterates (18) satisfies for all $t \geq 0$:*

$$L(w_{t+1}) \leq L(w_t) - \gamma (1 - \rho \beta) \Big( 1 - \frac{\gamma \beta}{2} (1 - \rho \beta) \Big) \|\nabla L(w_t)\|^2.$$

*If in addition the function $L$ satisfies (A3) with potentially $\mu = 0$, then for all $\gamma, \rho \geq 0$ such that $\gamma \beta (2 - \rho \beta) \leq 2$, we have*

$$L(w_{t+1}) \leq L(w_t) - \gamma \Big( 1 - \frac{\gamma \beta}{2} + \rho \mu \big( 1 - \gamma \beta - \frac{\gamma \rho \beta^2}{2} \big) \Big) \|\nabla L(w_t)\|^2.$$

We note that the constraints on the step-size are different depending on the assumptions on the function $L$. In the non-convex case, $\rho$ has to be smaller than $1/\beta$, whereas in the convex case, it has to be smaller than $2/\beta$.

*Proof.* Let us define by $w_{t+1/2} = w_t + \rho \nabla L(w_t)$ the SAM ascent step. Using the smoothness of the function $L$ (Assumption **(A2')**) (see, e.g., Nesterov, 2004), we obtain

$$L(w_{t+1}) \leq L(w_t) - \gamma \langle \nabla L(w_{t+1/2}), \nabla L(w_t) \rangle + \frac{\gamma^2 \beta}{2} \|\nabla L(w_{t+1/2})\|^2.$$

The main trick is to use the binomial squares

$$\|\nabla L(w_{t+1/2})\|^2 = -\|\nabla L(w_t)\|^2 + \|\nabla L(w_{t+1/2}) - \nabla L(w_t)\|^2 + 2\langle \nabla L(w_{t+1/2}), \nabla L(w_t) \rangle,$$

to bound

$$L(w_{t+1}) \leq L(w_t) - \gamma\langle\nabla L(w_{t+1/2}), \nabla L(w_t)\rangle + \frac{\gamma^2\beta}{2}\|\nabla L(w_{t+1/2})\|^2$$

$$= L(w_t) - \frac{\gamma^2\beta}{2}\|\nabla L(w_t)\|^2 + \frac{\gamma^2\beta}{2}\|\nabla L(w_{t+1/2}) - \nabla L(w_t)\|^2 - \gamma(1-\gamma\beta)\langle\nabla L(w_{t+1/2}), \nabla L(w_t)\rangle$$

$$\leq L(w_t) - \gamma[1 - \rho\beta - \frac{\gamma\beta}{2}(1-\rho\beta)^2]\|\nabla L(w_t)\|^2,$$

where we have used Lemma 7 and that $\|\nabla L(w_{t+1/2}) - \nabla L(w_t)\|^2 \leq \beta^2\|w_{t+1/2} - w_t\|^2 \leq \beta^2\rho^2\|\nabla L(w_t)\|^2$.

If in addition the function $L$ is convex then we can use its co-coercivity (see, e.g., Nesterov, 2004) to bound $\|\nabla L(w_{t+1/2}) - \nabla L(w_t)\|^2 \leq \beta\langle\nabla L(w_{t+1/2}) - \nabla L(w_t), w_{t+1/2} - w_t\rangle$ and obtain a tighter bound:

$$L(w_{t+1}) \leq L(w_t) - \gamma\langle\nabla L(w_{t+1/2}), \nabla L(w_t)\rangle + \frac{\gamma^2\beta}{2}\|\nabla L(w_{t+1/2})\|^2$$

$$= L(w_t) - \frac{\gamma^2\beta}{2}\|\nabla L(w_t)\|^2 + \frac{\gamma^2\beta}{2}\|\nabla L(w_{t+1/2}) - \nabla L(w_t)\|^2 - \gamma(1-\gamma\beta)\langle\nabla L(w_{t+1/2}), \nabla L(w_t)\rangle$$

$$\leq L(w_t) - \gamma(1 - \frac{\gamma\beta}{2})\|\nabla L(w_t)\|^2 - \gamma(1 - \gamma\beta - \frac{\gamma\rho\beta^2}{2})\langle\nabla L(w_{t+1/2}) - \nabla L(w_t), \nabla L(w_t)\rangle$$

$$\leq L(w_t) - \gamma(1 - \frac{\gamma\beta}{2} + \rho\mu(1 - \gamma\beta - \frac{\gamma\rho\beta^2}{2}))\|\nabla L(w_t)\|^2,$$

where we have used Lemma 7.  □

**Convergence proofs.** Using the previous Lemma 8 recursively, we can bound the average gradient value of the iterates (18) of SAM algorithm and ensure convergence to stationary points.

**Proposition 9.** *Assume (A2'). For any $\gamma < 1/\beta$ and $\rho < 1/\beta$, the iterates equation 18 satisfies for all $T \geq 0$:*

$$\frac{1}{T}\sum_{t=0}^{T}\|\nabla L(w_t)\|^2 \leq \frac{L(w_0) - L(w_T)}{T\gamma(1-\rho\beta)[1 - \frac{\gamma\beta}{2}(1-\rho\beta)]}.$$

*Proof.* Using the Lemma 8 we obtain

$$\gamma(1-\rho\beta)\Big(1 - \frac{\gamma\beta}{2}(1-\rho\beta)\Big)\|\nabla L(w_t)\|^2 \leq L(w_t) - L(w_{t+1}).$$

And summing these inequalities for $t = 0, \cdots, T-1$ yields

$$\frac{1}{T}\sum_{t=0}^{T-1}\|\nabla L(w_t)\|^2 \leq \frac{L(w_0) - L(w_T)}{T\gamma(1-\rho\beta)[1 - \frac{\gamma\beta}{2}(1-\rho\beta)]}.$$

□

When the function $L$ additionally satisfies a Polyak-Lojasiewicz condition, i.e., the Assumption **(A3)**, linear convergence of the function value to the minimum function value can be obtained. This is the object of the following proposition:

**Proposition 10.** *Assume (A2'-3). For any $\gamma < 1/\beta$ and $\rho < 1/\beta$, the iterates (18) satisfies for all $T \geq 0$:*

$$L(w_t) - L_* \leq \Big(1 - 2\gamma\mu(1-\rho\beta)\Big(1 - \frac{\gamma\beta}{2}(1-\rho\beta)\Big)\Big)^t (L(w_0) - L_*).$$

*Proof.* Using the Lemma 8 and that the function $L$ is $\mu$ Polyak-Lojasiewicz (Assumption **(A3)**) we obtain

$$L(w_{t+1}) \leq L(w_t) - 2\mu\gamma(1-\rho L)\Big(1 - \frac{\gamma\beta}{2}(1-\rho L)\Big)(L(w_t) - L_*).$$

And subtracting the optimal value $L_*$ we get

$$L(w_t) - L_* \leq \left(1 - 2\gamma\mu(1 - \rho\beta)\left(1 - \frac{\gamma\beta}{2}(1 - \rho\beta)\right)\right)(L(w_{t-1}) - L_*)$$

$$\leq \left(1 - 2\gamma\mu(1 - \rho\beta)\left(1 - \frac{\gamma\beta}{2}(1 - \rho\beta)\right)\right)^t (L(w_0) - L_*).$$

$\square$

When the function $L$ is convex, convergence of the average of the iterates can be proved.

**Proposition 11.** *Assume (A2') and $L$ convex. For any step-sizes $\gamma$ and $\rho$ such that $\gamma\beta(1 + \rho\beta) < 2$, then the averaged $\bar{w}_T = \frac{1}{T}\sum_{t=0}^{T-1} w_t$ of the iterates (18) satisfies for all $T \geq 0$:*

$$L(\bar{w}_T) - L_* \leq \frac{2\rho\beta + 1}{\gamma(2 - \gamma\beta(1 + \rho\beta))T}\|w_0 - w_*\|^2,$$

*If, in addition, the function $L$ is $\mu$-strongly convex, then:*

$$\|w_T - w_*\|^2 \leq \left(1 - \gamma\mu(2 - \gamma\beta(1 + \rho\beta))\right)^T (2\rho + 1)\|w_0 - w_*\|^2.$$

The proof is using a different astute Lyapunov function (which works for the non-strongly convex case).

*Proof.* Let us define by $V_t = [L(w_t) - L(w_*)] + \frac{1}{2\rho}\|w_t - w_*\|^2$ and by $w_{t+1/2} = w_t + \rho\nabla L(w_t)$ the SAM ascent step.

$$V_{t+1} - V_t \leq -\frac{\gamma}{\rho}\langle\nabla L(w_{t+1/2}), w_t - w_*\rangle - \gamma\langle\nabla L(w_{t+1/2}), \nabla L(w_t)\rangle + \frac{\gamma^2}{2\rho}(1 + \rho\beta)\|\nabla L(w_{t+1/2})\|^2$$

$$= -\frac{\gamma}{\rho}\langle\nabla L(w_{t+1/2}), w_t + \rho\nabla L(w_t) - w_*\rangle + \frac{\gamma^2}{2\rho}(1 + \rho\beta)\|\nabla L(w_{t+1/2})\|^2$$

$$= -\frac{\gamma}{\rho}\langle\nabla L(w_{t+1/2}), w_{t+1/2} - w_*\rangle + \frac{\gamma^2}{2\rho}(1 + \rho\beta)\|\nabla L(w_{t+1/2})\|^2$$

$$\leq -\frac{\gamma}{\rho}(1 - \frac{\gamma\beta}{2}(1 + \rho\beta))\langle\nabla L(w_{t+1/2}), w_{t+1/2} - w_*\rangle.$$

If $L$ is convex then $L(w_{t+1/2}) - L(w_*) \leq \langle\nabla L(w_{t+1/2}), w_{t+1/2} - w_*\rangle$ and therefore we obtain

$$\frac{\gamma}{\rho}(1 - \frac{\gamma\beta}{2}(1 + \rho\beta))L(w_{t+1/2}) - L(w_*) \leq V_t - V_{t+1}.$$

Using the definition of $w_{t+1/2}$ we always have that $L(w_{t+1/2}) \geq L(w_t) + \rho\|\nabla L(w_t)\|^2$ therefore

$$\frac{\gamma}{\rho}(1 - \frac{\gamma\beta}{2}(1 + \rho\beta))L(w_t) - L(w_*) \leq V_t - V_{t+1}.$$

And taking the sum and using Jensen inequality we finally obtain:

$$L(1/T\sum_{t=0}^{T} w_t) - L(w_*) \leq \frac{V_0 - V_{T+1}}{T\frac{\gamma}{\rho}(1 - \frac{\gamma\beta}{2}(1 + \rho\beta))}.$$

If $L$ is $\mu$-strongly convex, we use that $\langle\nabla L(w_{t+1/2}), w_{t+1/2} - w_*\rangle \geq \mu\|w_{t+1/2} - w_*\|^2$ to obtain

$$\|w_{t+1/2} - w_*\|^2 = \|w_t + \rho\nabla L(w_t) - w_*\|^2 = \|w_t - w_*\|^2 + 2\rho\langle\nabla L(w_t), w_t - w_*\rangle + \rho^2\|\nabla L(w_t)\|^2$$

$$\geq \|w_t - w_*\|^2 + 2\rho\langle\nabla L(w_t), w_t - w_*\rangle$$

$$\geq \|w_t - w_*\|^2 + 2\rho[L(w_t) - L(w_*)]$$

$$\geq 2\rho V_t.$$

Therefore we have

$$V_{t+1} \leq (1 - \gamma\mu(2 - \gamma\beta(1 + \rho\beta))V_t \leq (1 - \gamma\mu(2 - \gamma\beta(1 + \rho\beta))^{t+1}V_0.$$

$\square$

## B.2 CONVERGENCE OF STOCHASTIC IMPLEMENTATIONS OF SAM ALGORITHM

### B.2.1 CONVERGENCE OF MAXSUM GRADIENT ALGORITHM

When the SAM algorithm is implemented with the MaxSum objective as optimization objective, two different batches are using in the ascent and descent steps. We obtain the MaxSum algorithm defined as

$$w_{t+1} = w_t - \frac{\gamma_t}{b} \sum_{i \in I_t} \nabla \ell_i \big( w_t + \frac{\rho_t}{b} \sum_{i \in J_t} \nabla \ell_i(w_t) \big), \tag{19}$$

where $I_t$ and $J_t$ are two different mini-batches of data of size $b$. For this variant of the algorithm we obtain a similar convergence result that the one obtain with the same batch for the two steps.

**Proposition 12.** *Assume (A1), (A2') for the iterates (19). For any $T \geq 0$ and for step-sizes $\gamma_t = \frac{1}{\sqrt{T}\beta}$ and $\rho_t = \frac{1}{T^{1/4}\beta}$, we have:*

$$\frac{1}{T} \mathbb{E} \sum_{t=0}^{T-1} \|\nabla L(w_t)\|^2] \leq \frac{4}{\beta\sqrt{T}}(L(w_0) - L_*) + \frac{8\sigma^2}{b\sqrt{T}},$$

*In addition, under (A2), with step-sizes $\gamma_t = \min\{\frac{8t+4}{3\mu(t+1)^2}, \frac{1}{2\beta}\}$ and $\rho_t = \sqrt{\gamma_t/\beta}$:*

$$\mathbb{E}[L(w_T)] - L_* \leq \frac{3\beta^2(L(w_0) - L_*)}{\mu^2 T^2} + \frac{22\beta\sigma^2}{b\mu^2 T}$$

We obtain the same convergence result that in Proposition 2, but under the relaxed smoothness assumption **(A2')**.

As in the deterministic case, the proof relies on two lemmas which shows that the SAM update is well correlated with the gradient and that the decrease of function values can be controlled.

**Auxilliary Lemmas.** The following lemma shows that the SAM update is well correlated with the gradient $\nabla L(w_t)$. Let us denote by $\nabla L_{t+1}(w) = \frac{1}{b} \sum_{i \in I_t} \nabla \ell_i(w)$, $\nabla L_{t+1/2}(w) = \frac{1}{b} \sum_{i \in J_t} \nabla \ell_i(w)$ and $w_{t+1/2} = w_t + \rho \nabla L_{t+1/2}(w_t)$ the SAM ascent step.

**Lemma 13.** *Assume (A1) and (A2). Then for all $\rho \geq 0$, $t \geq 0$ and $w \in \mathbb{R}^d$,*

$$\mathbb{E}\langle \nabla L_{t+1}(w + \rho \nabla L_{t+1/2}(w)), \nabla L(w) \rangle \geq (1/2 - \beta\rho)\|\nabla L(w)\|^2 - \frac{\beta^2 \rho^2 \sigma^2}{2}.$$

The proof is similar to the proof of Lemma 7. Only the stochasticity of the noisy gradients has to be taken into account. In this aim, we consider instead the update which would have been obtained without noise, and bound the remainder using the bounded variance assumption **(A1)**.

*Proof.* Let us denote by $\hat{w} = w + \rho \nabla L(w)$, the true gradient step. We first add and subtract $\nabla L_{t+1/2}(\hat{w})$

$$\langle \nabla L_{t+1}(w + \rho \nabla L_{t+1/2}(w)), \nabla L(w) \rangle = \langle \nabla L_{t+1}(w + \rho \nabla L_{t+1/2}(w)) - \nabla L_{t+1}(\hat{w}), \nabla L(w) \rangle - \langle \nabla L_{t+1}(\hat{w}), \nabla L(w) \rangle.$$

We bound the two terms separately. We use the smoothness of $L$ (Assumption **(A2')**) to bound the first term:

$$-\mathbb{E}\langle \nabla L_{t+1}(w + \rho \nabla L_{t+1/2}(w)) - \nabla L_{t+1}(\hat{w}), \nabla L(w) \rangle = -\mathbb{E}\langle \nabla L(w + \rho \nabla L_{t+1/2}(w)) - \nabla L(\hat{w}), \nabla L(w) \rangle$$

$$\leq \frac{1}{2} \mathbb{E} \|\nabla L(w + \rho \nabla L_{t+1/2}(w)) - \nabla L(\hat{w})\|^2 + \frac{1}{2}\|\nabla L(w)\|^2$$

$$\leq \frac{\beta^2}{2} \mathbb{E} \|w + \rho \nabla L_{t+1/2}(w) - \hat{w}\|^2 + \frac{1}{2}\|\nabla L(w)\|^2$$

$$\leq \frac{\beta^2 \rho^2}{2} \mathbb{E} \|\nabla L_{t+1/2}(w) - \nabla L(w)\|^2 + \frac{1}{2}\|\nabla L(w_t)\|^2$$

$$\leq \frac{\beta^2 \rho^2 \sigma^2}{2b} + \frac{1}{2} \|\nabla L(w)\|^2,$$

where we have used that the variance of a mini-batch of size $b$ is bounded by $\sigma^2/b$. Note that this term can be equivalently bounded by $\beta\rho\sigma/\sqrt{b}\|\nabla L(w)\|$ if needed. For the second term, we directly apply Lemma 7 to obtain

$$\mathbb{E}\langle\nabla L_{t+1}(\hat{w}),\nabla L(w)\rangle = \mathbb{E}\langle\nabla L(\hat{w}),\nabla L(w)\rangle$$
$$\geq (1-\beta\rho)\|\nabla L(w)\|^2.$$

$\square$

The next lemma shows that the decrease of function values of SN-SAM algorithm can be controlled similarly as in the case of stochastic gradient descent.

**Lemma 14.** *Let us assume (A1, A2') then for all $\gamma \leq 1/(2\beta)$ and $\rho \leq 1/(2\beta)$, the iterates equation 19 satisfies*

$$\mathbb{E}\,L(w_{t+1}) \leq \mathbb{E}\,L(w_t) - \gamma/4\,\mathbb{E}\,\|\nabla L(w_t)\|^2 + \gamma\beta\sigma^2(\gamma+\rho^2 beta).$$

This lemma is analogous to Lemma 8 in the stochastic case. The proof is very similar, with the slight difference that Lemma 13 is used instead of Lemma 7.

*Proof.* Let us define by $w_{t+1/2} = w_t + \rho\nabla L_{t+1/2}(w_t)$. Using the smoothness of the function $L$ **(A2)**, we obtain

$$L(w_{t+1}) \leq L(w_t) - \gamma\langle\nabla L_{t+1}(w_{t+1/2}),\nabla L(w_t)\rangle + \frac{\gamma^2\beta}{2}\|\nabla L_{t+1}(w_{t+1/2})\|^2.$$

Taking the expectation and using that the variance is bounded **(A1)** yields to

$$\mathbb{E}\,L(w_{t+1}) \leq \mathbb{E}\,L(w_t) - \gamma\,\mathbb{E}\langle\nabla L(w_{t+1/2}),\nabla L(w_t)\rangle + \frac{\gamma^2\beta}{2}\,\mathbb{E}\,\|\nabla L_{t+1}(w_{t+1/2})\|^2$$
$$\leq \mathbb{E}\,L(w_t) - \gamma\,\mathbb{E}\langle\nabla L(w_{t+1/2}),\nabla L(w_t)\rangle + \gamma^2\beta\,\mathbb{E}\,\|\nabla L_{t+1}(w_{t+1/2}) - \nabla L(w_{t+1/2})\|^2 + \gamma^2\beta\,\mathbb{E}\,\|\nabla L(w_{t+1/2})\|^2$$
$$\leq \mathbb{E}\,L(w_t) - \gamma\,\mathbb{E}\langle\nabla L(w_{t+1/2}),\nabla L(w_t)\rangle + \gamma^2\beta\sigma^2/b + \gamma^2\beta\,\mathbb{E}\,\|\nabla L(w_{t+1/2})\|^2.$$

The main trick is still to use the binomial squares

$$\|\nabla L(w_{t+1/2})\|^2 = -\|\nabla L(w_t)\|^2 + \|\nabla L(w_{t+1/2}) - \nabla L(w_t)\|^2 + 2\langle\nabla L(w_{t+1/2}),\nabla L(w_t)\rangle$$

to bound

$$\mathbb{E}\,L(w_{t+1}) \leq \mathbb{E}\,L(w_t) - \gamma\,\mathbb{E}\langle\nabla L(w_{t+1/2}),\nabla L(w_t)\rangle + \frac{\gamma^2\beta}{2}\,\mathbb{E}\,\|\nabla L(w_{t+1/2})\|^2 + \gamma^2\sigma^2\beta/b$$
$$= \mathbb{E}\,L(w_t) - \gamma^2 L\,\mathbb{E}\,\|\nabla L(w_t)\|^2 + \gamma^2\beta\,\mathbb{E}\,\|\nabla L(w_{t+1/2}) - \nabla L(w_t)\|^2$$
$$\quad - \gamma(1-2\gamma\beta)\,\mathbb{E}\langle\nabla L(w_{t+1/2}),\nabla L(w_t)\rangle + \gamma^2\sigma^2\beta/b$$
$$= \mathbb{E}\,L(w_t) - \gamma^2\beta\,\mathbb{E}\,\|\nabla L(w_t)\|^2 + \gamma^2 L^3\,\mathbb{E}\,\|w_{t+1/2} - w_t\|^2$$
$$\quad - \gamma(1-2\gamma\beta)(1/2+\alpha\rho)\,\mathbb{E}\,\|\nabla L(w_t)\|^2 + \gamma(1-2\gamma L)\sigma^2\rho^2\beta^2/2 + \gamma^2\sigma^2\beta/b$$
$$= \mathbb{E}\,L(w_t) - \gamma^2\beta\,\mathbb{E}\,\|\nabla L(w_t)\|^2 + \gamma^2\beta^3\rho^2\,\mathbb{E}\,\|\nabla L_{t+1/2}(w_t)\|^2$$
$$\quad - \gamma(1-2\gamma\beta)(1/2+\alpha\rho)\,\mathbb{E}\,\|\nabla L(w_t)\|^2 + \gamma(1-2\gamma\beta)\sigma^2/b\rho^2\beta^2/2 + \gamma^2\sigma^2\beta/b$$
$$= \mathbb{E}\,L(w_t) - \gamma^2\beta\,\mathbb{E}\,\|\nabla L(w_t)\|^2 + 2\gamma^2\beta^3\rho^2\,\mathbb{E}\,\|\nabla L(w_t)\|^2 + 2\gamma^2\beta^3\rho^2\sigma^2/b$$
$$\quad - \gamma(1-2\gamma\beta)(1/2+\alpha\rho)\,\mathbb{E}\,\|\nabla L(w_t)\|^2 + \gamma(1-2\gamma\beta)\sigma^2\rho^2\beta^2/2 + \gamma^2\sigma^2\beta/b$$
$$\leq L(w_t) - \frac{\gamma}{2}[1 - 2\rho\beta(1-2\gamma\beta(1-\rho\beta))]\,\mathbb{E}\,\|\nabla L(w_t)\|^2 + \gamma\sigma^2\beta/b[\gamma + \rho^2 L/2(1+2\gamma\beta)]$$

where we have used Lemma 13 and that $\|\nabla L(w_{t+1/2}) - \nabla L(w_t)\|^2 \leq \beta^2\|w_{t+1/2} - w_t\|^2$. $\square$

Using Lemma 14 we directly obtain the following convergence result.

**Proposition 15.** *Assume (A1) and (A2'). For $\gamma \leq 1/(2\beta)$ and $\rho \leq 1/(2\beta)$, the iterates equation 5 satisfies:*

$$\frac{1}{T}\sum_{t=0}^{T-1}\mathbb{E}\|\nabla L(w_t)\|^2 \leq 4\frac{L(w_0) - \mathbb{E}\,L(w_T)}{T\gamma} + 4T\sigma^2\beta(\gamma + \rho^2\beta)/b.$$

This proposition gives the first part of Proposition 12. The proof of the stronger result obtained when the function is in addition PL (Assumption **(A3)**) is similar to the proof of Theorem 3.2 of Gower et al. (2019), only the constants are changing.

### B.2.2 CONVERGENCE OF SUMMAX GRADIENT ALGORITHM

When the SAM algorithm is implemented to minimize the SumMax objective, the same batch is used in the ascent and descent steps. We obtain then iterates (5) for which we have stated the convergence result in Proposition 2. The proof follows the same lines as before with the minor different that we are assuming the individual gradient $\nabla f_t$ are Lipschitz (Assumption **(A2)**) to control the alignment of the expected SAM direction. Let us denote by $\nabla L_t(w) = \frac{1}{b}\sum_{i\in J_t}\nabla\ell_i(w)$.

**Lemma 16.** *Assume (A1-2). Then we have for all $w \in \mathbb{R}^d$, $\rho \geq 0$ and $t \geq 0$*

$$\mathbb{E}\langle\nabla L_t(w + \rho\nabla L_t(w)), \nabla L(w)\rangle \geq (1/2 - \rho\beta)\|\nabla L(w)\|^2 - \frac{\beta^2\rho^2\sigma^2}{2b}.$$

The proof is very similar to the proof of Lemma 13. The only difference is that the Assumption **(A2)** is used instead of **(A2')**.

*Proof.* Let us denote by $\hat{w} = w + \rho\nabla L(w)$, the true gradient step. We first add and subtract $\nabla L_t(\hat{w})$

$$\langle\nabla L_t(w + \rho\nabla L_t(w)), \nabla L(w)\rangle = \langle\nabla L_t(w + \rho\nabla L_t(w)) - \nabla L_t(\hat{w}), \nabla L(w)\rangle - \langle\nabla L_t(\hat{w}), \nabla L(w)\rangle.$$

We bound the two terms separately. We use the smoothness of $L_t$ to bound the first term (Assumption **(A2)**):

$$-\langle\nabla L_t(w + \rho\nabla L_t(w)) - \nabla L_t(\hat{w}), \nabla L(w)\rangle \leq \frac{1}{2}\|\nabla L_t(w + \rho\nabla L_t(w)) - \nabla L_t(\hat{w})\|^2 + \frac{1}{2}\|\nabla L(w)\|^2$$

$$\leq \frac{\beta^2}{2}\mathbb{E}\,\|w + \rho\nabla L_t(w) - \hat{w}\|^2 + \frac{1}{2}\|\nabla L(w)\|^2$$

$$\leq \frac{\beta^2\rho^2}{2}\|\nabla L_t(w) - \nabla L(w)\|^2 + \frac{1}{2}\|\nabla L(w)\|^2.$$

And taking the expectation, we obtain:

$$-\mathbb{E}\langle\nabla L_t(w + \rho\nabla L_t(w)) - \nabla L_t(\hat{w}), \nabla L(w)\rangle \leq \frac{\beta^2\rho^2\sigma^2}{2b} + \frac{1}{2}\mathbb{E}\,\|\nabla L(w)\|^2.$$

For the second term, we apply directly Lemma 7

$$\mathbb{E}\langle\nabla L_t(\hat{w}), \nabla L(w_t)\rangle = \langle\nabla L(\hat{w}), \nabla L(w)\rangle$$

$$\geq (1 - \beta\rho)\|\nabla L(w)\|^2.$$

Assembling the two inequalities yields to the result. $\square$

The next lemma shows that the decrease of function values of SAM algorithm can be controlled similarly as in the case of gradient descent. It is analogous to Lemma 8 where different batches are used in both the ascent and descent steps of SAM algorithm.

**Lemma 17.** *Assume (A1-2). For all $\gamma \leq 1/\beta$ and $\rho \leq 1/(4\beta)$, the iterates (5) satisfies*

$$\mathbb{E}\,L(w_{t+1}) \leq \mathbb{E}\,L(w_t) - 3\gamma/8\,\mathbb{E}\,\|\nabla L(w_t)\|^2 + \gamma\beta\sigma^2/b(\gamma + 2\rho^2\beta).$$

*Proof.* Let us define by $w_{t+1/2} = w_t + \rho \nabla L_{t+1}(w_t)$. Using the smoothness of the function $L$ which is implied by **(A2)**, we obtain

$$L(w_{t+1}) \le L(w_t) - \gamma \langle \nabla L_{t+1}(w_{t+1/2}), \nabla L(w_t) \rangle + \frac{\gamma^2 \beta}{2} \|\nabla L_{t+1}(w_{t+1/2})\|^2.$$

We still use the binomial squares

$$\|\nabla L_{t+1}(w_{t+1/2})\|^2 = -\|\nabla L(w_t)\|^2 + \|\nabla L_{t+1}(w_{t+1/2}) - \nabla L(w_t)\|^2 + 2\langle \nabla L_{t+1}(w_{t+1/2}), \nabla L(w_t) \rangle$$

and bound $L(w_{t+1})$ by

$$L(w_{t+1}) \le L(w_t) - \frac{\gamma^2 \beta}{2}\|\nabla L(w_t)\|^2 + \frac{\gamma^2 \beta}{2}\|\nabla L_{t+1}(w_{t+1/2}) - \nabla L(w_t)\|^2 - \gamma(1 - \gamma\beta)\langle \nabla L_{t+1}(w_{t+1/2}), \nabla L(w_t) \rangle$$

$$\le L(w_t) - \frac{\gamma^2 \beta}{2}\|\nabla L(w_t)\|^2 + \gamma^2 \beta\|\nabla L_{t+1}(w_{t+1/2}) - \nabla L_{t+1}(w_t)\|^2 + \gamma^2 \beta\|\nabla L_{t+1}(w_t) - \nabla L(w_t)\|^2$$
$$- \gamma(1 - \gamma\beta)\langle \nabla L_{t+1}(w_{t+1/2}), \nabla L(w_t) \rangle$$

$$\le L(w_t) - \frac{\gamma^2 \beta}{2}\|\nabla L(w_t)\|^2 + \gamma^2 \beta\beta^2\|w_{t+1/2} - w_t\|^2 + \gamma^2 \beta\|\nabla L_{t+1}(w_t) - \nabla L(w_t)\|^2$$
$$- \gamma(1 - \gamma\beta)\langle \nabla L_{t+1}(w_{t+1/2}), \nabla L(w_t) \rangle$$

$$= L(w_t) - \frac{\gamma^2 \beta}{2}\|\nabla L(w_t)\|^2 + \gamma^2 \beta^3 \rho^2\|\nabla L_{t+1}(w_t)\|^2 + \gamma^2 \beta\|\nabla L_{t+1}(w_t) - \nabla L(w_t)\|^2$$
$$- \gamma(1 - \gamma\beta)\langle \nabla L_{t+1}(w_{t+1/2}), \nabla L(w_t) \rangle$$

$$= L(w_t) - \frac{\gamma^2 \beta}{2}(1 - 4\beta^2 \rho^2)\|\nabla L(w_t)\|^2 + \gamma^2 \beta(1 + 2\beta^2 \rho^2)\|\nabla L_{t+1}(w_t) - \nabla L(w_t)\|^2$$
$$- \gamma(1 - \gamma\beta)\langle \nabla L_{t+1}(w_{t+1/2}), \nabla L(w_t) \rangle$$

Taking the expectation and using Lemma 16 we obtain

$$\mathbb{E}\, L(w_{t+1}) \le \mathbb{E}\, L(w_t) - \frac{\gamma^2 \beta}{2}(1 - 4\beta^2 \rho^2)\,\mathbb{E}\,\|\nabla L(w_t)\|^2 + \gamma^2 \beta(1 + 2\beta^2 \rho^2)\,\mathbb{E}\,\|\nabla L_{t+1}(w_t) - \nabla L(w_t)\|^2$$
$$- \gamma(1 - \gamma\beta)\,\mathbb{E}\langle \nabla L_{t+1}(w_{t+1/2}), \nabla L(w_t) \rangle$$

$$\le \mathbb{E}\, L(w_t) - \frac{\gamma^2 \beta}{2}(1 - 4\beta^2 \rho^2)\,\mathbb{E}\,\|\nabla L(w_t)\|^2 + \gamma^2 \beta(1 + 2\beta^2 \rho^2)\sigma^2/b$$
$$- \gamma(1 - \gamma\beta)(1/2 - \beta\rho)\,\mathbb{E}\,\|\nabla L(w_t)\|^2 + \gamma(1 - \gamma\beta)\frac{\rho^2 \sigma^2 \beta^2}{2b}$$

$$\le \mathbb{E}\, L(w_t) - \frac{\gamma^2 \beta}{2}(1 - 4\beta^2 \rho^2)\,\mathbb{E}\,\|\nabla L(w_t)\|^2 + \gamma^2 \beta(1 + 2\beta^2 \rho^2)\sigma^2/b$$
$$- \frac{\gamma}{2}(1 - 2\beta\rho(1 - \gamma(\beta - 2\rho\beta^2)))\,\mathbb{E}\,\|\nabla L(w_t)\|^2 + \gamma\sigma^2/b[\gamma\beta + \frac{\rho^2 \beta^2}{2}(1 + 3\gamma\beta)].$$

□

Using Lemma 17 we directly obtain the following convergence result.

**Proposition 18.** *Assume (A1-2). For $\gamma \le 1/\beta$ and $\rho \le 1/4\beta$, the iterates (5) satisfies:*

$$\frac{1}{T}\sum_{t=0}^{T-1} \mathbb{E}\,\|\nabla L(w_t)\|^2 \le 8/3\frac{L(w_0) - \mathbb{E}\,L(w_T)}{T\gamma} + 8T\sigma^2 \beta(\gamma + \rho^2 \beta)/(3b).$$

*In addition, under (A3), with step-sizes $\gamma_t = \min\{\frac{8t+4}{3\mu(t+1)^2}, 1/(2\beta)\}$ and $\rho_t = \sqrt{\gamma_t/\beta}$:*

$$\mathbb{E}[L(w_T)] - L_* \le \frac{3\beta^2(L(w_0) - L_*)}{\mu^2 T^2} + \frac{22\beta\sigma^2}{\mu^2 bT}.$$

*Proof.* The first bound directly comes from Lemma 17. The second bound is similar to the proof of Theorem 3.2 of Gower et al. (2019), only the constants are changing.  □

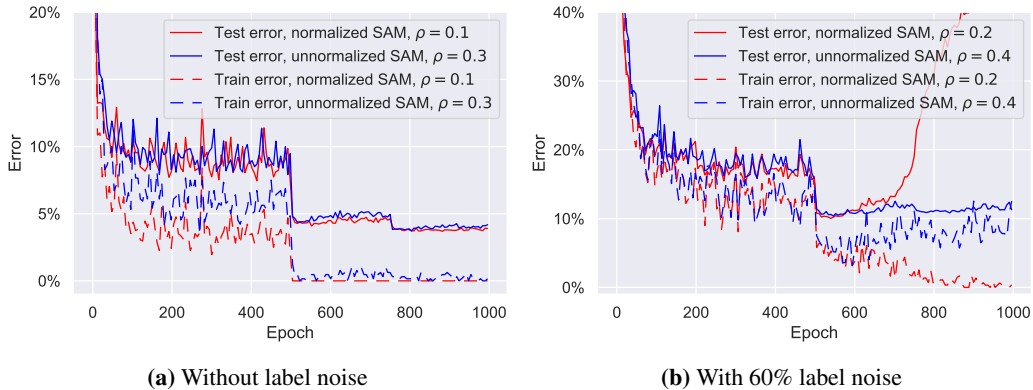

**(a)** Without label noise            **(b)** With 60% label noise

**Figure 10:** Training and test error over training epochs for the normalized and unnormalized formulations of SAM. We can see that both variants of SAM improve the test error to a similar level.

### B.3    GRADIENT NORMALIZATION IN SAM

We consider here the implementation of SAM algorithm where the inner gradient step is normalized:

$$w_{t+1} = w_t - \gamma \nabla L\Big(w_t + \rho \frac{\nabla L(w_t)}{\|\nabla L(w_t)\|}\Big), \tag{20}$$

We state here several claims showing that the normalized formulation in Eq.(20) can be less favorable than the unnormalized one of Eq.(18).

**Fact I: Normalized SAM does not converge with constant step-size.** The last iterate of SAM with constant step-size is never converging which can be easily seen on the function $L(w) = 0.5w^2$ for $w \in \mathbb{R}$. Indeed SAM iterates equation 20 become $w_{t+1} = (1 - \gamma)w_t - \gamma\rho \operatorname{sign}(w_t)$, and convergence implies that the limit $w_\infty$ would satisfy the fixed point $w_\infty = -\rho \operatorname{sign}(w_\infty)$ which never holds for $\rho > 0$.

**Fact II: Normalized SAM when averaged does not necessarily converge to a flatter minimum.** Let us consider the following simple function

$$L(w) = \begin{cases} 1/2w^2 & \text{if } w \geq 0 \\ 1/6w^6 & \text{if } w < 0. \end{cases}$$

When applied to this function with $\gamma = \rho = 1$, starting from $w_0 = -1$, then it is experimentally observed that the averaged of the iterates (20) converge to a positive point $w_\infty = 0.0149$ which is sharper than the origin, i.e, $S_L^1(w_\infty) = 0.515 > S_L^1(0) = 0.5$.

**Fact III: Normalized SAM can stay stuck at non-stationary points of the function it optimizes.** For any $\rho$, there exists a non-convex function $L_\rho$ and a point $w\rho$ such that $\nabla L_\rho(w_\rho + \rho \frac{\nabla L_\rho(w_\rho)}{\|\nabla L_\rho(w_\rho)\|}) = 0$ and $\nabla L_\rho(w_\rho) \neq 0$. To see that, just consider the function $-0.5x^2$ and $w_\rho = \rho$.

**Performance of the unnormalized SAM for deep networks.** Given the potential shortcomings of the normalization step in SAM, it is interesting to compare the generalization properties of the normalized and unnormalized versions. For this, we repeat the experiments in Sec. 3.3 and Sec. 4 but for the unnormalized formulation in Fig. 10 with a grid search over $\rho$. We can observe that although the optimal $\rho$ is not the same for both formulations ($\rho = 0.3$ vs $\rho = 0.1$ without label noise and $\rho = 0.4$ vs $\rho = 0.2$ with 60% label noise), the test error of the normalized vs unnormalized variants is very similar: 3.69% vs 3.70% without label noise, and 10.03% vs 10.29% with 60% label noise. Thus, given the same generalization performance, the unnormalized version may be preferred due to the lack of the shortcomings described above.

## C   EXPERIMENTAL DETAILS

**Experiments on deep networks.**   In all experiments, we train deep networks using SGD with step size 0.1 and momentum 0.9, and $\ell_2$-regularization parameter $\lambda = 0.0005$. We use three datasets: CIFAR-10 Krizhevsky & Hinton (2009), CIFAR-100 Krizhevsky & Hinton (2009), and two-class CIFAR-10 where we select two random classes (*horse* and *car*). We train models with basic data augmentations: random image crops and mirroring. We use a pre-activation ResNet-18 (He et al., 2016) with a width factor 64 and piece-wise constant learning rates (with a 10-times decay at 50% and 75% epochs) for all experiments except Table 2 where we follow the setup of (Huang et al., 2020) and use ResNet-34 with cosine learning rates with one cycle. We train all models for 1000 epochs except the experiments in Tables 2, and Fig. 13 for which we use 200 epochs as they require more expensive training (due to adversarial training) or an expensive grid search over each dataset and label noise amount.

For all experiments involving SAM, we select the best $\rho$ based on a grid search over $\rho \in \{0.025, 0.05, 0.1, 0.2, 0.3, 0.4\}$. In the fine-tuning ERM with SAM experiment, we used $\rho = 0.2$. We do not use the $m$-sharpness (Foret et al., 2021) since we were not able to reproduce exactly the results of the SAM paper with batch size of 256 and $m = 32$ (as suggested by Foret et al. (2021)) for the noisy label experiments. Instead, we observed that the same performance can be achieved using a smaller batch size 128 and $m = 128$, thus we opted for this setting which has an advantage of being significantly faster (thus, we could perform a detailed grid search for each setting) if only one GPU is available per run.

For model selection to determine the best epoch early stopping and the best $\rho$, we use 10% of examples from the *noisy* training set as the validation set following (Zhang & Sabuncu, 2018). We emphasize that we *do not* assume access to a *clean* validation set to perform early stopping.

For stochastic weight averaging (SWA), we use an exponential moving average with parameter $\tau = 0.999$, and we update the moving average every *iteration*. Moreover, we start SWA from the very beginning of training (e.g., similarly to Rebuffi et al. (2021)) which is not the same setting as proposed in Izmailov et al. (2018) who start SWA from a specific epoch, use a different moving average coefficient, and a modified learning rate schedule. We deviate from the setting of Izmailov et al. (2018) since it has much more hyperparameters than the single moving average parameter $\tau$, and those hyperparameters have to be chosen differently for the noisy label setting compared to the standard setting of Izmailov et al. (2018).

For experiments with the generalized cross-entropy loss (GCE) Zhang & Sabuncu (2018) we use $q = 0.7$ as recommended in Zhang & Sabuncu (2018) unless mentioned otherwise.

For experiments with $\ell_\infty$ adversarial training, we use $\epsilon = 8/255$ as in Madry et al. (2018); Rice et al. (2020), 5 iterations of Projected Gradient Descent (PGD) for training and 10 iterations for testing (following Rice et al. (2020)), step size $2\epsilon/n_{steps}$ for training (as in Madry et al. (2018)) and step size $\epsilon/4$ for evaluation (as in Croce & Hein (2020)). Note that we use PGD with 10 iterations to get an understanding of the qualitative behavior over epochs without using expensive computations. However, to strengthen the evaluation, we also use AutoAttack Croce & Hein (2020) to assess the robustness of the best model over training. The model selection is performed via the validation robust error obtained via 10-iteration PGD.

**Experiments on a linear model.**   Here we provide details for the experiments shown in Fig. 4. We train a logistic regression model without regularization using full-batch gradient descent. We use 50 training examples which are sampled from two Gaussians in a 100-dimensional space. Each Gaussian has the standard deviation $\sigma = 0.1$, and the mean of each Gaussian $\mu$ is sampled randomly from the unit sphere. For optimization, we initialize the weight vector randomly on the sphere of radius $1/3$ and use 50 iterations of gradient descent with a step size 2.0. We use $\rho = 5.0$ for SAM.

**Computing infrastructure.**   We perform all our experiments with deep networks on NVIDIA V100 GPUs with 32GB of memory.

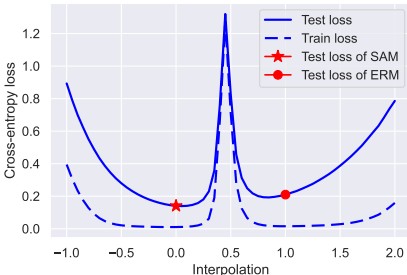 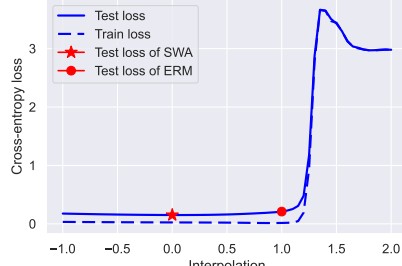

**Figure 11:** Loss interpolations of $w_{SAM}$ vs $w_{ERM}$ and $w_{SWA}$ vs $w_{ERM}$. We see that the loss interpolation for SAM is qualitatively different from SWA.

## D    ADDITIONAL EXPERIMENTS

**Differences between SAM and Stochastic Weight Averaging.** Stochastic Weight Averaging (SWA) (Izmailov et al., 2018) is a weight averaging technique that improves generalization and, similarly to SAM, also motivated from the perspective of *sharpness* of the loss, thus it is natural to ask whether SAM shares some similarities with SWA. To study this, we plot the loss interpolations between the weights of SAM, ERM, and SWA models ($w_{SAM}$, $w_{ERM}$, $w_{SAM}$) in Fig. 11 which suggest that SAM and ERM converge to different minima. This is further supported by the fact that $\|w_{SAM} - w_{SWA}\|_2$ is large, i.e., $\approx 1/3$ of the distance of $w_{SAM}$ to its initialization. Interestingly, there is a peak between $w_{SAM}$ and $w_{ERM}$ which is, however, not too high and the model at the peak still has non-trivial test error (around 40%). This is in contrast to $w_{SWA}$ which is located in the same valley as $w_{ERM}$ with a very sharp increase close to $w_{erm}$ (similarly to the interpolations from Izmailov et al. (2018)). We also note another difference between SAM and SWA in the context of the fine-tuning experiment from Fig. 2: fine-tuning an SWA model will lead to slight overfitting as it would converge to a standard ERM model (Izmailov et al., 2018) contrary to SAM which improves generalization.

**Does the loss derivative change explain the behavior of a non-linear model?** Here we provide more details on the experiment from Sec. 4 about the importance of changing the loss derivative $\ell'(y_i \cdot f_{x_i}(w))$ with SAM as opposed to the direction of the individual gradients $\nabla f_{x_i}(w)$.

We train a pre-activation ResNet-18 (He et al., 2016) with a width factor 16 (instead of the standard 64) on a two-class CIFAR-10 dataset (*horse* vs *car* classes) using batch size 256 and $m_{SAM} = 64$ which refers to the $m$-sharpness defined in Foret et al. (2021)). We note that this experiment would be more difficult to perform for the multi-class setting due to the fact that $\ell'(y_i \cdot f_{x_i}(w_{SAM}))$ will not be a scalar. In order to select $\rho$ for each of the SAM-based methods, we performed a grid search over $\{0.001, 0.003, 0.01, 0.03, 0.1, 0.2\}$. We train each model three times and report the average test error (obtained with early stopping on the validation set) with the standard deviation.

In Fig. 12, we plot the test error of all the four methods over epochs. We can see that although the test error for *SAM derivative* is oscillating (unlike for other methods), the best test error over epochs ($7.45\% \pm 0.28$) is lower than for other methods and close to that of SAM ($9.02\% \pm 0.34$). We emphasize that all reported results are obtained via proper model selection over epochs using a validation set. To conclude, we can see that by discarding the direction of SAM the test errors start to oscillate but we still preserve the improved generalization if we use early stopping.

**Beneficial effect of combining SAM with GCE.** We check how predictive is our explanation about the advantage of using SAM with a robust loss such as GCE (Zhang & Sabuncu, 2018) instead of CE to better leverage the early-learning phenomenon. We validate this hypothesis experimentally using a ResNet-34 network trained on CIFAR-10 and CIFAR-100 datasets and present results in Table 2 for $\{40\%, 60\%\}$ label noise (for the further details see App. C). We present results from the literature (Huang et al., 2020) where all of them use ResNet-34 except MentorNet which uses WRN-28-10. Early stopping based on a noisy validation set is done for each method except those reported from the literature. We can see that SAM indeed performs better with GCE than with CE which leads to competitive results. Moreover, the results can be further enhanced by using weight averaging (Izmailov et al., 2018) where it outperforms many recently proposed approaches including

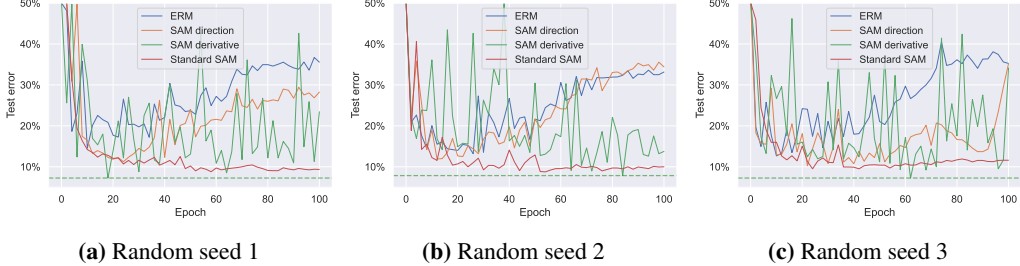

| **(a)** Random seed 1 | **(b)** Random seed 2 | **(c)** Random seed 3 |

**Figure 12:** Test error over epochs for *ERM*, *SAM direction*, *SAM derivative*, and *SAM* for three random seeds. Although the test error for *SAM derivative* is oscillating, the best test error over epochs (dashed line) is lower than for other methods.

**Table 2:** Test error of CE + SAM and GCE + SAM under label noise compared to methods from the literature. Using SAM with the GCE loss works better against label noise compared to using SAM with the CE loss as predicted by our gradient reweighting interpretation of SAM.

| | CIFAR-10 | | CIFAR-100 | |
|---|---|---|---|---|
| % LABEL NOISE | 40% | 60% | 40% | 60% |
| SCE (WANG ET AL., 2019) | 13.26% | 19.20% | 33.59% | 42.57% |
| MENTORNET (JIANG ET AL., 2018) | 11.00% | - | 32.00% | - |
| DAC (THULASIDASAN ET AL., 2019) | 9.29% | 13.70% | 33.08% | 42.83% |
| SELF-ADAPTIVE TRAINING (HUANG ET AL., 2020) | 7.36% | 10.77% | 28.62% | 37.31% |
| CE | 17.06% | 23.06% | 44.82% | 58.30% |
| CE + SAM | 7.29% | 10.98% | 36.79% | 43.02% |
| GCE + SAM | **7.18%** | 10.41% | 28.82% | 37.32% |
| GCE + SAM + WEIGHT AVERAGING | 7.62% | **9.74%** | **27.79%** | **35.91%** |

Self-Adaptive Training (Huang et al., 2020). We also note that it can be further combined with other successful approaches against label noise such as MixUp (Zhang et al., 2017), bootstrapping (Reed et al., 2014), and semi-supervised learning approaches (Li et al., 2020a).

**Mitigating robust overfitting.** To illustrate how techniques from label noise can be transferred to adversarial training and improve upon robust overfitting, we benchmark here two approaches: (1) generalized cross-entropy (GCE) loss and (2) semi-supervised pairing terms (Laine & Aila, 2016; Luo et al., 2019). We train all methods for 200 epochs as in Rice et al. (2020). To the best of our knowledge, these approaches have not been used to prevent robust overfitting.

In Fig. 13a, we show the performance of adversarial training with the GCE loss (we use $q = 0.9$) instead of the standard cross-entropy loss. We observe that GCE is able to mitigate robust overfitting, however it does not improve the performance of the best model over epochs compared to the cross-entropy loss, so its usefulness is limited.

In Fig. 13b, we show the results of adversarial training with an additional semi-supervised pairing term (Laine & Aila, 2016) that has been used to mitigate label noise in, e.g., Luo et al. (2019) (see also Li et al. (2020a) for a more advanced approach). Specifically, let $f$ be the logits of the network, $x_i^{adv}$ be the $\ell_\infty$ adversarial example generated for sample $x_i$ using the PGD attack and $\bar{x}_i$ be a version of $x_i$ with a different augmentation. Then the additional term that we use in the objective is:

$$\lambda \cdot \frac{1}{n} \sum_{i=1}^{n} \left\| f(x_i^{adv}) - f(\bar{x}_i) \right\|_2^2, \tag{21}$$

which we use together with the cross-entropy on adversarial examples as in standard adversarial training (Madry et al., 2018). We use $\lambda = 1.0$ and a "warmup" scheme for the regularizer which consists in starting the regularizer after 50% of training epochs which is a standard practice (Laine & Aila, 2016; Luo et al., 2019). Both $\lambda$ and the starting epoch were determined via a grid search. We show in Fig. 13b the robust error over epochs given by PGD with 10 iteration of adversarial training with and without the pairing term. We can observe that adding the pairing term mitigates the robust overfitting trend, *and* leads to improved robust error.

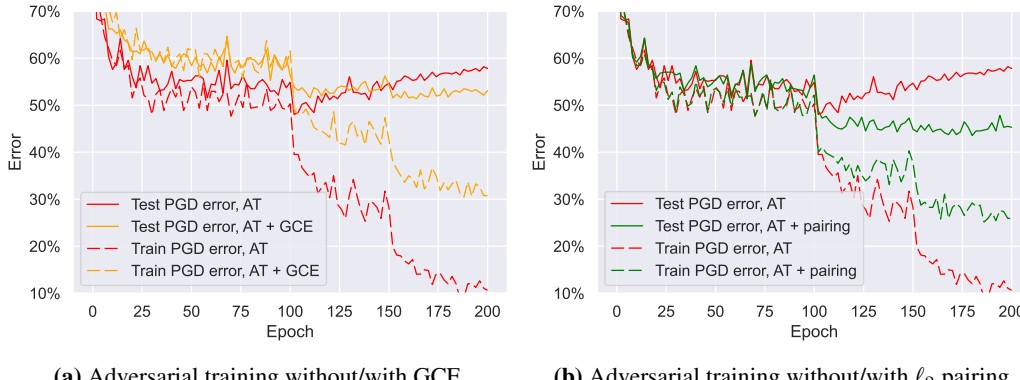

**(a)** Adversarial training without/with GCE

**(b)** Adversarial training without/with $\ell_2$ pairing

**Figure 13:** Robust error over epochs for standard adversarial training compared to adversarial training with GCE and with the $\ell_2$ pairing term.

To make sure that the improvement is not due to gradient masking (Carlini et al., 2019), we evaluate the best model additionally with AutoAttack (Croce & Hein, 2020). The model trained with the pairing term achieves 51.15% robust error while the standard adversarial training leads to 52.56% robust error and adversarial training with SAM achieves 50.07% robust error. Thus, we conclude that the techniques from the label noise literature (beyond simple early stopping (Rice et al., 2020)) can be successfully transferred to improve upon robust overfitting. Moreover, we note that more advanced pairing terms are possible, e.g. involving temperature scaling of the logits (Berthelot et al., 2019), more advanced data augmentations (Sohn et al., 2020), and other improvements over the standard pairing (Li et al., 2020a). We leave applications of more advanced pairing techniques to improve upon robust overfitting as future work.

Finally, the improvement from the pairing term may also be connected to the fact that the TRADES objective (Zhang et al., 2019) often works better than standard adversarial training (see Gowal et al. (2020) for an exhaustive empirical study). The approach of TRADES also adds an additional pairing term that does not rely on the ground truth labels to the standard cross-entropy loss (although TRADES uses a different loss and does not use a different augmentation on $\bar{x}_i$). This suggests that even some of the existing successful techniques can be interpreted based on the connection between label noise and robust overfitting.

