# OpenReview forum: "Understanding Sharpness-Aware Minimization"
_ICLR.cc/2022/Conference — ICLR 2022 Submitted_

### Official Review · Reviewer_iJV1 · 2021-10-29

**Correctness:** 4
**Technical Novelty And Significance:** 2
**Empirical Novelty And Significance:** 2
**Recommendation:** 3
**Confidence:** 3

**Main Review:**

This is an ambitious paper that seeks to cover many a lot of ground, from convergence properties of SAM, to generalization error, to robustness in noisy labels and adversarial training. The reviewer believes that each of these issues merits a single paper, so to squeeze all the discussions into one paper actually dilutes each of the contributions, as I will discuss below.

The authors did a nice job to introduce the MaxSum SAM formulation and stochastic gradient descent-type algorithms to optimize the minmax objective. The first major result is Proposition 2 in which the authors use analytical techniques from SGD to control the expected gradient of SAM with stochastic gradients. The problem with this result is that, while mathematically elegant, it does not explain why SAM outperforms SGD. A theoretical explanation of SAM must be accompanied by comparison of the theory to SAM.

The next contribution has to do with SAM's robustness to noisy labels. This is good aim to study. However, the explanation falls short of being convincing. The authors merely use the fact that SAM changes the direction of the gradient to be less aligned to the noisy direction. They substantiated this fact using Figs. 3b and 3c. But this is merely *one* experiment -- ResNet-18 trained on CIFAR-10 with 60% label noise. The reviewer was hoping for a more convincing theoretical explanation for this.

The third contribution is the observation that SAM even improves generalization for linear models. This is a surprising phenomenon, but the explanation is not clear. The authors substantiate this not through theory, but through a toy example in in Fig. 4. It is not clear whether one can extrapolate the observations from this linear model to large-scale datasets. Again, I was hoping for a more theoretical explanation, since this is an "Understanding" paper.

**Summary Of The Paper:**

This is an ambitious paper that seeks to explain why SAM works the way it does. First, the authors try to justify the better generalization of SAM over SGD on deep networks. Next, the authors discuss why SAM is robust to noisy labels, and argue why SAM can improve generalization even for linear classifiers. Finally, the authors discuss how to further improve SAM by leveraging a gradient reweighting interpretation and combining with a robust loss.

**Summary Of The Review:**

The aims of this paper are laudable, but the explanations for short of being convincing. I would suggest the authors, in a new version of the paper, to focus solely on one aspect of SAM, and explain it fully (using theoretical results and comprehensive experiments), and comparing it to vanilla methods.

---

> ### Author Response · Authors · 2021-11-24
> **We thank the reviewer for the useful feedback. We discuss the main points below.**
>
> **Organization of the paper**
>
> > *The reviewer believes that each of these issues merits a single paper, so to squeeze all the discussions into one paper actually dilutes each of the contributions*
>
> Our perspective was that it is important to explain the observed benefits of SAM in each setting: standard, noisy label, robust training. These settings are quite different: e.g., in the standard/noiseless setting, we can train the model to zero loss without any overfitting while in the noisy label setting, we always have to rely on early stopping (even when using SAM) which is the reason why we studied them separately.
>
> ---
>
> **Convergence results**
>
> > *The first major result is Proposition 2 in which the authors use analytical techniques from SGD to control the expected gradient of SAM with stochastic gradients. The problem with this result is that, while mathematically elegant, it does not explain why SAM outperforms SGD.*
>
> We agree that the convergence result on its own indeed doesn’t suggest any advantage of SAM over SGD. **However, this point is precisely addressed in Proposition 1 which shows the beneficial implicit bias of SumMax and MaxSum SAM over ERM.** The implicit bias result requires showing the convergence of the loss for SAM which is partially addressed in Proposition 2. In addition, we believe that showing convergence of a new algorithm, e.g., to stationary points in the non-convex setting, is the necessary first step of any theoretical understanding. And it’s worth noting that this result for SAM together with the precise conditions for the SAM step size were unknown before, even in the convex setting.
>
> ---
>
> **Noisy label results**
>
> > *The authors merely use the fact that SAM changes the direction of the gradient to be less aligned to the noisy direction. They substantiated this fact using Figs. 3b and 3c. But this is merely one experiment -- ResNet-18 trained on CIFAR-10 with 60% label noise. The reviewer was hoping for a more convincing theoretical explanation for this.*
>
> We will make sure to repeat this experiment on a different dataset and replicate it over multiple random seeds.
>
>
> > *The third contribution is the observation that SAM even improves generalization for linear models. This is a surprising phenomenon, but the explanation is not clear. The authors substantiate this not through theory, but through a toy example in in Fig. 4. It is not clear whether one can extrapolate the observations from this linear model to large-scale datasets.*
>
> First of all, we already know the observations regarding the performance of SAM on noisy labels for large-scale datasets/models due to the original SAM paper (Foret et al., 2021). The goal of the linear model experiment in our paper was rather to show a very simple setting where SAM still exhibits generalization benefits and then relate this setting to the behaviour of SAM on deep networks (i.e., Fig. 3 and also paragraph **“Does the loss derivative change explain the behavior of a non-linear model?”**). We agree that providing a formal analysis of the findings from Sec. 4 would be interesting (especially, as future work) but we also think that providing interpretations and hypotheses based on empirical observations (like in our Sec. 4) can still be valuable for a better understanding of SAM.

---

### Official Review · Reviewer_BAw4 · 2021-11-02

**Correctness:** 3
**Technical Novelty And Significance:** 1
**Empirical Novelty And Significance:** 2
**Recommendation:** 3
**Confidence:** 4

**Main Review:**

 I think this paper is well organized, and the proof seems solid. But I still have the following concerns:
1. This paper doesn't analyze SAM proposed in [1]. In the SAM algorithm, the gradient is normalized in the inner step. However, the algorithm proposed in this paper doesn't include such gradient norm terms. Just as the authors have summarized in the abstract, "$\ldots$ relies on worst-case weight perturbations," the gradient norm term is one of the most important parts of the SAM algorithm.

They constructed the counterexamples for the original SAM algorithm in section B.3 to show the strength of using an unnormalized one. However, those examples are far from the practice because they are strongly convex with global minima at point zero. In this case, it is obvious that the algorithm will never converge to zero, even with a small perturbation. In practice, however, we use the cross-entropy loss for classification problems in which the good local minima would be far from zero. The training loss also has many local minima. In this case, a small perturbation will still guarantee the convergence and help find the flat minima.

In order to show the algorithm proposed in this paper outperforms the original SAM algorithm, it would be more convincing if the authors could add more empirical comparison. I also want to ask whether the experiments in sections 3.3, 4, and 5 follow the SAM algorithm without normalization.

2. I get quite confused about whether MaxSum is better than SumMax. Before the implicit bias part on page 3, the authors write, "We next show formally that MaxSum SAM has a better implicit bias than ERM and SumMax SAM." Then, in the empirical part on page 4, the authors conclude that "ERM and MaxSum SAM enjoy the same performance whereas SumMax benefits from a better implicit bias."

3. Most of the results in this paper implicitly use the square loss rather than cross-entropy loss which limits the impact of the results. Does the implicit bias exist for cross-entropy loss? If so, it is worth adding it to the paper.

4. The convergence results in section 3.2 are based on assumptions A1 and A2. Generally, do you think assumptions A1 and A2 hold for NNs? Or do those assumptions hold for the diagonal linear neural networks analyzed in section 3.1?

5. Figure 1 shows that ERM and MaxSum SAM enjoy the same performance. I think maybe $\rho$ is too small, have the authors tuned the parameter of $\rho$?

[1] Foret, Pierre, et al. "Sharpness-aware Minimization for Efficiently Improving Generalization." International Conference on Learning Representations. 2020.

**Summary Of The Paper:**

In this paper, the authors mainly analyzed a SAM-like algorithm with a square loss. For diagonal linear networks, this paper proves that the NNs trained by the algorithm has better generalizations than the ones trained by normal SGD. For deep NNs satisfies PL condition and other assumptions mentioned in section3, the authors can prove the algorithm will converge. The authors use a linear model to give the intuition why the SAM algorithm can generalize better on a dataset with label noise.

**Summary Of The Review:**

Is the main contribution of this paper to understand the original SAM algorithm or to propose a new SAM-like algorithm with a theoretical guarantee? I think that the paper has to make more clear what are the main contributions. The proof technique used in this paper is quite standard, and the results are based on the square loss. So the impact of the results may be limited. I also have some concerns about some details, as I have mentioned in the main review. So currently, I would like to suggest a rejection, but I am open to discussion and willing to increase my score.

---

> ### Author Response · Authors · 2021-11-24
> **We thank for the useful feedback. We discuss below the point about unnormalized SAM and other points (Part 2)**
>
> **Square loss**
>
> > *Most of the results in this paper implicitly use the square loss rather than cross-entropy loss which limits the impact of the results. Does the implicit bias exist for cross-entropy loss? If so, it is worth adding it to the paper.*
>
> We would like to clarify that we use the square loss *only in Sec. 3.1* for the formal study of the implicit bias of SAM. Regarding the implicit bias for cross-entropy loss, showing a difference in the implicit bias for MaxSum / SumMax SAM will be more subtle (for ERM, see [Moroshko et al. (2020)](https://arxiv.org/abs/2007.06738)) as all these methods are expected to converge to the minimum $||\beta||_1$ solution for diagonal linear networks. Thus, finding an appropriate model to formally understand the implicit bias of SAM in the classification setting is an interesting direction for future work.
>
> ---
>
> **Convergence analysis**
>
> > *The convergence results in section 3.2 are based on assumptions A1 and A2. Generally, do you think assumptions A1 and A2 hold for NNs? Or do those assumptions hold for the diagonal linear neural networks analyzed in section 3.1?*
>
> Both assumptions A1 (bounded variance of stochastic gradients) and A2 (individual $\beta$-smoothness) should hold for neural networks (including diagonal linear ones from Sec. 3.1) with smooth losses (such as cross-entropy and square losses). The assumption A2 (even without stochastic gradients) requires the inputs $x$ to be bounded in norm, i.e. $||x||_2 \leq C$ but this is typically satisfied (e.g., images are all in $x \in [0, 1]^d$ and thus are bounded).

---

> ### Author Response · Authors · 2021-11-24
> **We thank for the useful feedback. We discuss below the point about unnormalized SAM and other points (Part 1)**
>
> **The performance of unnormalized SAM**
>
> > *This paper doesn't analyze SAM proposed in [1]. In the SAM algorithm, the gradient is normalized in the inner step. However, the algorithm proposed in this paper doesn't include such gradient norm terms.*
>
> Indeed, we analyze a version of SAM in Eq. 5 that relies on step sizes $\rho_t$ that do not depend on the gradient norm. However, we do show in Appendix B.3 that **the generalization benefits of SAM with and without normalization are empirically the same** (see Fig. 10), both without and with label noise. In particular, the normalized vs. unnormalized SAM achieve 3.69% vs 3.70% test error without label noise, and 10.03% vs. 10.29% test error with 60% label noise.
>
>
> > *the gradient norm term is one of the most important parts of the SAM algorithm*
>
> Could we kindly ask the reviewer to point out some empirical evidence that can support this claim? As a side note, when we compared the unnormalized and normalized SAM, we found it important to do a grid search for the inner step size $\rho$ as typically a several times larger $\rho$ is required for the unnormalized SAM.
>
>
> > *In order to show the algorithm proposed in this paper outperforms the original SAM algorithm, it would be more convincing if the authors could add more empirical comparison.*
>
> We would like to clarify that we do not make strong claims about the superiority of the unnormalized SAM. Instead, we just point out several facts in Appendix B.3 that are based on our convergence analysis that can be of interest to better understand this subtlety of SAM.
>
>
>
> > *I also want to ask whether the experiments in sections 3.3, 4, and 5 follow the SAM algorithm without normalization.*
>
> All the experiments with SAM in our paper were performed **with normalization** except the experiments in Appendix B.3. We agree that the submitted paper did not clearly mention this, so we have updated the paper to clarify this (paragraph “SAM with normalized gradient”, Sec. 3.2).
>
>
> > *Is the main contribution of this paper to understand the original SAM algorithm or to propose a new SAM-like algorithm with a theoretical guarantee? I think that the paper has to make more clear what are the main contributions.*
>
> Our main contribution is to provide a better understanding of the **original SAM** and we think that a short discussion of the normalization in SAM can be beneficial for that. We have updated the text to briefly reflect this.
>
> ---
>
> **SumMax is better than MaxSum**
>
> > *I get quite confused about whether MaxSum is better than SumMax. Before the implicit bias part on page 3, the authors write, "We next show formally that MaxSum SAM has a better implicit bias than ERM and SumMax SAM." Then, in the empirical part on page 4, the authors conclude that "ERM and MaxSum SAM enjoy the same performance whereas SumMax benefits from a better implicit bias."*
>
> We regret that we made this typo. The first quoted sentence should instead be:
>
> *"We next show formally that **SumMax** SAM has a better implicit bias than ERM and **MaxSum** SAM."*
>
> We have fixed it in the updated version of the paper.
>
>
> > *Figure 1 shows that ERM and MaxSum SAM enjoy the same performance. I think maybe $\rho$ is too small, have the authors tuned the parameter of $\rho$?*
>
> For Figure 1, we used a fixed $\rho$ which was the same for both MaxSum SAM and SumMax SAM. Tuning $\rho$ for each method separately can help to achieve a better test loss for both methods ($\rho_0$ denotes the inner step size of SAM we used in the paper for Fig. 1):
>
> | $\beta$ | Test loss MaxSum | Test loss SumMax |
> | ------------- |-------------| -----|
> | $\beta_0$ | 0.0012 |  0.0002 |
> | $2 \cdot \beta_0$ | 0.0011 |  0.0000075 |
> | $4 \cdot \beta_0$ | 0.0011 |  0.0000000011 |
> | $8 \cdot \beta_0$ | 0.0009  |  1.0000 |
> | $16 \cdot \beta_0$ | 0.0003 | 1.0000 |
> | $32 \cdot \beta_0$ | 0.0132 | 1.0000 |
> | $64 \cdot \beta_0$ | 1.7340 |  1.0000 |
>
> However, the performance of SumMax SAM remains significantly better even after a grid search over $\rho$.

---

### Official Review · Reviewer_EJMz · 2021-11-03

**Correctness:** 3
**Technical Novelty And Significance:** 3
**Empirical Novelty And Significance:** 3
**Recommendation:** 6
**Confidence:** 4

**Main Review:**

### Strengths
The main strength of the paper is the theoretically sound and insightful analysis of SAM.

### Weakness
* In its present state, the paper unfortunately reads a bit like a collection of disparate specialized results, and there is no deep fundamental insight gained why SAM improves generalization over SGD.  Perhaps this could be addressed by providing a more speculative / high-level discussion in the conclusion / outlook section.
* It is proven that SAM converges to a stationary point of the original loss (similar to SGD). To me, this is actually a weakness of SAM and not a strength, since ideally one wants a robust method to converge to points in the landscape which are non-stationary but in an overall "flat" low-loss region.
* In the paper, it is mentioned that the averaged iterates of normalized-SAM converge to a different point and it is argued that this "different" point is a bad one ("not necessarily flatter than the solution").  Is this conclusion based on some empirical evidence or a theoretical insight? It would be great if the paper could elaborate on this point.
* Should Foret et al (2021) and Wu et al (2020) really be considered concurrent work? They are quite far apart (Apr2020 vs Oct2020 on arXiv).
* Also, there is the concurrent work to SAM https://arxiv.org/abs/2010.04925 which should be mentioned.

**Summary Of The Paper:**

The aim of the paper is to provide theoretical explanations for the recent successes of adversarial weight perturbation and sharpness-aware minimization (SAM) methods. In particular, these methods have been shown to significantly help robust generalization in deep learning and theoretical explanations for their success are missing so far.

The paper aims to fill this gap by getting a better understanding of SAM. It contributes a collection of miscellaneous results:
* A proof that SAM provides better generalization for linear neural networks than SGD
* The convergence of SAM is analyzed and generalization behavior is discussed
* A new interpretation of SAM is presented in terms of gradient reweighing
* Finally, a connection of SAM to the noisy label literature is made

**Summary Of The Review:**

I believe that the collection of theoretical results on SAM is both novel and useful to the community. Moreover, the paper is well-written. If the issues mentioned in my main review can be addressed, I am inclined to increase my score and recommend acceptance of the paper.

---

> ### Author Response · Authors · 2021-11-24
> **Thanks for the detailed review. We address below the main points.**
>
> **Understanding SAM as a unifying theme**
> > *In its present state, the paper unfortunately reads a bit like a collection of disparate specialized results, and there is no deep fundamental insight gained why SAM improves generalization over SGD.*
>
>
> The unifying theme of the paper is providing a better understanding of different aspects of the SAM algorithm, both of its MaxSum and SumMax versions. All sections consider this question but from different perspectives:
> * Sec. 3.1 formally analyzes the implicit bias in a diagonal neural network,
> * Sec. 3.2 presents a formal convergence result which is necessary for studying the implicit bias,
> * Sec. 3.3 discusses convergence and generalization properties of SAM and when its effect is beneficial during training of deep networks,
> * Sec. 4 relies on empirical observations to provide an interpretation of the benefits of SAM in the noisy label setting,
> * Sec. 5 discusses the connection between adversarial training and label noise and conjectures why the effect of SAM (or Adversarial Weight Perturbation per [Wu et al. (2020)](https://arxiv.org/abs/2004.05884)) can be beneficial in the robust training setting.
>
> We think that the impression that our results are disparate may come from our separate treatment of the standard (Sec. 3) and noisy label (Sec. 4) settings. We believe that these two settings are quite different:
> * in the standard/noiseless setting (common for deep learning), we can train the model to zero loss without any overfitting (e.g., as Fig. 2 illustrates),
> * in the noisy label setting, we always have to rely on early stopping, even when using SAM.
>
>
> These differences between the two settings motivated us to study them independently.
>
> > Perhaps this could be addressed by providing a more speculative / high-level discussion in the conclusion / outlook section.
>
>
> We thank the reviewer for this suggestion. We will make sure to provide a clearer and more unifying discussion behind our results in the standard and noisy label settings.
>
> ---
>
> **A clarification regarding the convergence result**
> > *It is proven that SAM converges to a stationary point of the original loss (similar to SGD). To me, this is actually a weakness of SAM and not a strength, since ideally one wants a robust method to converge to points in the landscape which are non-stationary but in an overall "flat" low-loss region.*
>
> We would like to emphasize that our theoretical convergence results agree with what is observed in practice in the deep learning setting (i.e., Sec. 3.3): SAM achieves a nearly zero training loss (0.0009 for SAM vs. 0.0012 for ERM after 1000 epochs on CIFAR-10). I.e., SAM converges very close to the global minimum of the training objective. We believe that this makes our convergence results relevant for understanding SAM.
>
> Regarding convergence to non-stationary points, we are not sure what would be the advantage of converging to them as opposed to standard stationary points. However, we agree that intuitively converging to flat low-loss *regions* is an appealing idea.
>
>
>
> > *In the paper, it is mentioned that the averaged iterates of normalized-SAM converge to a different point and it is argued that this "different" point is a bad one ("not necessarily flatter than the solution"). Is this conclusion based on some empirical evidence or a theoretical insight? It would be great if the paper could elaborate on this point.*
>
> It is based on our short observation briefly discussed as Fact II in Appendix B.3. It shows an example of an asymmetric convex function where averaged normalized SAM empirically converges to a solution which is less flat than the optimum at 0. However, we note that although the normalization in SAM brings several concerns (formulated as Facts I, II, III in Appendix B.3), generalization benefits with and without normalization appear to be the same on standard deep learning benchmarks (see Figure 10).
>
> ---
>
>
> **Related works**
> > *Should Foret et al (2021) and Wu et al (2020) really be considered concurrent work? They are quite far apart (Apr2020 vs Oct2020 on arXiv).*
>
> We decided to consider them as *concurrent* works since the paper of Wu et al. (2020) wasn’t published at the time of Foret et al. (2020) was released and its published version had some changes compared to its version available in April 2020 (e.g., PAC-Bayesian analysis was added later). But this discussion is definitely not central to our paper so we have removed all the mentions of “*concurrent*” in the context of these two papers.
>
>
> > *Also, there is the concurrent work to SAM https://arxiv.org/abs/2010.04925 which should be mentioned.*
>
>
> We thank the reviewer for bringing up this reference. We have added a short discussion on the work of Zheng et al. (CVPR 2021) to the related work section.

---

> > ### Comment · Reviewer_EJMz · 2021-11-24
> > **Thanks for the clarifications!**
> >
> > > Regarding convergence to non-stationary points, we are not sure what would be the advantage of converging to them as opposed to standard stationary points. However, we agree that intuitively converging to flat low-loss regions is an appealing idea.
> >
> > Imagine a simple one-dimensional loss where the only stationary point is close to a "wall" in the loss landscape, i.e., the loss radically increases when one moves away from the stationary point to the left. Further assume that when moving to the right, the loss only increases slowly. A non-stationary point has the advantage that it can still have low loss, but be further away from that wall.
> >
> > > We would like to emphasize that our theoretical convergence results agree with what is observed in practice in the deep learning setting (i.e., Sec. 3.3): SAM achieves a nearly zero training loss (0.0009 for SAM vs. 0.0012 for ERM after 1000 epochs on CIFAR-10). I.e., SAM converges very close to the global minimum of the training objective. We believe that this makes our convergence results relevant for understanding SAM.
> >
> > I don't doubt the correctness of the theoretical convergence proof but was simply remarking that perhaps convergence to an optimal point of the training objective should not be sold as a good thing (see the above example).
> >
> > > It is based on our short observation briefly discussed as Fact II in Appendix B.3. It shows an example of an asymmetric convex function where averaged normalized SAM empirically converges to a solution which is less flat than the optimum at 0.
> >
> > Thanks for pointing to the example -- should it be read as (1/2) w^2 and (1/6) w^6  or 1/(2w^2) and 1/(6w^6)? Assuming the first one, I don't think that the point found by normalized SAM is actually a bad one in this example. In fact, this example explains what I meant with "finding a non-stationary point that is better than a stationary one".  Because similar to the example mentioned above, the w^6 function is a very steep wall where the loss radically increases, a robust model should be far away from that wall, which normalized+averaged SAM appears to find a bit.  Could you perhaps remind the reader at that point to the definitions of S_L^1? I was looking for it now, and was not able to find it quickly.
> >
> > > However, we note that although the normalization in SAM brings several concerns (formulated as Facts I, II, III in Appendix B.3), generalization benefits with and without normalization appear to be the same on standard deep learning benchmarks (see Figure 10).
> > I agree that non-convergence of SAM is an issue (Fact I), but Fact III for me is not a problem since in my opinion we do not want stationary points of the original loss but rather points which give low loss even when perturbed.
> >
> > It will be indeed a very intriguing finding if it turns out that the normalization in SAM is not needed! Perhaps this should be studied in a more exhaustive setting, i.e., reproducing some of the benchmark numbers from the original SAM paper without normalization.

---

> > > ### Author Response · Authors · 2021-11-25
> > > **Follow-up comments on the one-dimensional example and SAM vs. SWA**
> > >
> > > > *Imagine a simple one-dimensional loss where the only stationary point is close to a "wall" in the loss landscape*
> > >
> > > > *perhaps convergence to an optimal point of the training objective should not be sold as a good thing*
> > >
> > > We thank the reviewer for this great clarification. It is interesting to note that this example resembles the justification of the [Stochastic Weight Averaging method](https://arxiv.org/abs/1803.05407) where averaging leads to a solution which is further away from the sharper “wall” of the loss landscape and has better generalization properties. We totally agree that this example makes a lot of sense and it was in fact our first intuition about SAM. However, we found out that this intuition doesn’t directly apply for SAM as we discuss in Appendix D: **“Differences between SAM and Stochastic Weight Averaging”** based on one-dimensional loss surface interpolations (Figure 11).
> > >
> > > Moreover, both ERM and SAM converge very close to stationary points if we consider their gradient norms: $1.2 \cdot 10^{-4}$ for ERM and $8.1 \cdot 10^{-5}$ for SAM on CIFAR-10. This further suggests that while the idea of converging to some non-stationary point is interesting in general, there seems to be no indication that SAM does that more than ERM.
> > >
> > >
> > >
> > > > *Thanks for pointing to the example -- should it be read as (1/2) w^2 and (1/6) w^6 or 1/(2w^2) and 1/(6w^6)? Assuming the first one, …*
> > >
> > > Yes, that’s right, it was assumed to be read as (1/2) w^2 and (1/6) w^6.
> > >
> > >
> > >
> > > > *Could you perhaps remind the reader at that point to the definitions of $S_L^1$?*
> > >
> > > By $S_L^\rho(w)$ we meant sharpness defined as in [Foret et al. (2021)](https://arxiv.org/abs/2010.01412): $S_L^\rho(w) = max_{||\varepsilon||_2 \leq \rho} L(w+\varepsilon) - L(w)$. So $S_L^1$ stands for sharpness of loss $L$ taken at radius $\rho=1$.
> > >
> > >
> > > > *Assuming the first one, I don't think that the point found by normalized SAM is actually a bad one in this example. In fact, this example explains what I meant with "finding a non-stationary point that is better than a stationary one". Because similar to the example mentioned above, the w^6 function is a very steep wall where the loss radically increases, a robust model should be far away from that wall, which normalized+averaged SAM appears to find a bit.*
> > >
> > > Actually, the function $\frac{1}{6} w^6$ grows very slowly compared to $\frac{1}{2} w^2$ in the vicinity of the minimum (i.e., $|w| \leq 1$). So then in this example the “steep wall” will be rather the positive part of the function, i.e. $\frac{1}{2} w^2$, which will grow significantly faster close to the origin. Our point was to show a simple example for which the averaged normalized SAM converges to a point $w_\infty=0.0149$ with a *higher sharpness* at the scale $\rho=1$ than the global minimum $w_{min}=0$.
> > >
> > >
> > > > *It will be indeed a very intriguing finding if it turns out that the normalization in SAM is not needed! Perhaps this should be studied in a more exhaustive setting, i.e., reproducing some of the benchmark numbers from the original SAM paper without normalization.*
> > >
> > > We agree with this and we will provide more experiments with unnormalized SAM.

---

> > > > ### Author Response · Authors · 2021-11-25
> > > > **Additional results for unnormalized SAM**
> > > >
> > > > Here are additional results for **unnormalized SAM** for a ResNet-18 trained on **CIFAR-100** using the same hyperparameters as in Figure 10 for CIFAR-10. We report the test error with early stopping based on a validation set:
> > > >
> > > >
> > > > **Standard CIFAR-100**
> > > >
> > > > | Method | Test error |
> > > > | -------- |-------|
> > > > | ERM | 23.3% |
> > > > | Normalized SAM, $\rho=0.1$ | 21.3% |
> > > > | Unnormalized SAM, $\rho=0.3$ | 21.3% |
> > > >
> > > > **CIFAR-100 with 60% label noise**
> > > >
> > > >
> > > > | Method | Test error |
> > > > | -------- |-------|
> > > > | ERM | 48.6% |
> > > > | Normalized SAM, $\rho=0.2$ | 42.9% |
> > > > | Unnormalized SAM, $\rho=0.4$ | 39.1% |
> > > >
> > > >
> > > > We can see that both normalized and unnormalized SAM give a substantial improvement over ERM also on CIFAR-100.

---

### Official Review · Reviewer_nmv6 · 2021-11-09

**Correctness:** 4
**Technical Novelty And Significance:** 3
**Empirical Novelty And Significance:** Not applicable
**Recommendation:** 3
**Confidence:** 3

**Main Review:**

This work provides an analysis of the sharpness aware minimization (SAM) by Foret et al. 2021. While the paper has some experiments, they are minimal and serve to reinforce the points made by the theoretical analyses. Therefore, I view this work to be a primarily theoretical work.

The first result uses the technique of Woodworth et al. 2020 to analyze the implicit bias of SAM when applied to 2-layer diagonal linear network. Although the model considered is very limited, and although the analysis technique is not novel, I still think this is a good first step. The authors summarize the results of their analysis in Proposition 1, but I am not sure if I agree with the interpretation of the anlayses. I do see that the implicit objectives differ by a factor of \alpha^(1/n) but this is a mere multiplicative factor that should not overall affect the meaning of the implicit optimization problem. In particular, I don't fully understand the reasoning behind the claim "The SumMax implementations has better bias properties since its effective scale of α is considerably smaller than the one of MaxSum." I would be happy to be corrected in the rebuttal, but otherwise I feel that Proposition 1 does not provide much insight into why MaxSum, SumMax, and regular SGD have different generalization performances.

The second result is a convergence analysis of SAM. This is a result that was missing in the original SAM paper, so it's nice that the authors establish that SAM converges to a stationary point, just as SGD does. However, the analytical techniques are not quite novel, and the conclusion says nothing about why SAM is better than SGD.

Finally, the authors provide discussions on why SAM provides benefits when labels are noisy and how SAM prevents robust overfitting. While the discussions here are interesting, they are not rigorous mathematical discussions, and the experimental validation of these discussions is not very thorough. Therefore, I view the contribution of these sections to be minimal.

**Summary Of The Paper:**

This work provides an analysis of the sharpness aware minimization (SAM) by Foret et al. 2021.

**Summary Of The Review:**

Overall, the paper presents some interesting and useful analyses, but the analytical techniques are not very novel and they do not (at least to me) provide significant clarity on why SAM outperforms SGD.

---

> ### Author Response · Authors · 2021-11-24
> **Thanks for the feedback. We provide more explanations about the implicit bias part and other points raised.**
>
> **General comment**
>
> > *While the paper has some experiments, they are minimal and serve to reinforce the points made by the theoretical analyses. Therefore, I view this work to be a primarily theoretical work.*
>
> We think that our paper should not be seen as exclusively theoretical. We tried to use all available tools to provide a better understanding of the benefits of SAM, including both formal analysis (the implicit bias and convergence parts) and empirical observations (the label noise and robust overfitting parts).
>
> ---
>
> **Implicit bias of SAM**
>
> > *I do see that the implicit objectives differ by a factor of $\alpha^{1/n}$ but this is a mere multiplicative factor that should not overall affect the meaning of the implicit optimization problem*
>
> We would like to clarify that the differing factor $\approx 1/n$ appears in the exponent inside of the $\alpha_{MaxSum}$ term which leads to a different value of $\alpha$ in the potential function $\phi_\alpha(\beta) = \sum_{i=1}^d \alpha^2_i q(\beta_i/\alpha^2_i)$, where $q(z) = 2-\sqrt{4+z^2}+z\text{ arcsinh}(z/2)$. This effectively leads to **two different functions of $\beta$ that SumMax SAM and MaxSum SAM implicitly minimize**: $\phi_{\alpha_{SumMax}}(\beta)$ and $\phi_{\alpha_{MaxSum}}(\beta)$. **The meaning of the implicit bias then also changes**: SumMax SAM implicitly minimizes a function which is closer to $||\beta||_1$ compared to MaxSum SAM (see, e.g., Theorem 2 in [Woodworth et al. (2020)](https://arxiv.org/abs/2002.09277) for a precise bound on $||\beta||_1$ depending on $\alpha$).
>
> It can be also helpful to compare our results with the work of [Woodworth et al. (2020)](https://arxiv.org/abs/2002.09277) where they showed that for diagonal neural networks, the *scale of the initialization* $\alpha$ used for gradient descent controls the transition between the “kernel” and “rich” regimes, i.e., between minimum $\ell_1$-norm and $\ell_2$-norm solutions (in terms of $\beta$) for this model. Our analysis suggests that SumMax SAM can also aid this transition via a much smaller parameter $\alpha_{SumMax}$ (for a fixed initialization scale) that change the implicitly minimized norm to be *closer* to the $\ell_1$ norm rather than $\ell_2$ norm. The numerical experiments (Figure 1) confirm that the implicit bias of SumMax SAM indeed brings generalization benefits for a sparse regression problem.
>
>
> > *although the analysis technique is not novel*
>
> We believe that the proof technique itself doesn’t necessarily have to be novel to get some new insights for an empirically successful algorithm. In particular, we believe that our result is interesting because (1) it’s the first result on the implicit bias of SAM, (2) it doesn’t rely on *sharpness*, thus providing a complementary view to the original SAM paper.
>
> ---
>
> **Convergence of SAM**
>
> > *the conclusion [from the convergence result] says nothing about why SAM is better than SGD*
>
> The convergence result on its own indeed doesn’t suggest any advantage of SAM over SGD. However, our main motivation to establish the convergence result comes from considering the implicit bias of SAM which requires showing the convergence of the loss. In addition, we believe that showing convergence of a new algorithm, e.g., to stationary points in the non-convex setting, is the necessary first step of any theoretical understanding. And it’s worth noting that this result for SAM together with the precise conditions for the SAM step size were unknown before, even in the convex setting.
>
> ---
>
> **Effect of SAM for noisy labels**
>
> > *While these discussions [about label noise] are interesting, they are not rigorous mathematical discussions*
>
> The discussions in the label noise part indeed do not contain formal results and are based on empirical observations. However, we think that they are still valuable as *hypotheses* which can be interesting to validate theoretically in future work. We think that the gradient reweighting interpretation of SAM and its relation to robust losses is an interesting perspective on SAM which is complementary to the implicit bias view of SAM presented in Sec. 3.
>
>
> > *the experimental validation of these discussions is not very thorough.*
>
> We would appreciate it if the reviewer could point out specific experiments that would be worth validating further to support our arguments. We would be happy to replicate them on other datasets/models.

---

### Decision · Program_Chairs · 2022-01-20

**Decision:**

Reject

**Comment:**

As evident by the title the paper focuses on understanding sharpness-aware minimization which is a contemporary training procedure  based on minimizing the worse case perturbation of the weights in ball. It has been observed that SAM improves the generalization and this paper aims to demystify this success. They also provide a convergence proof of SAM for non-convex objectives in a simplified setting and also discuss benefits of SAM in the noisy label setting.The reviewers thought this paper was an interesting first step The reviewers raised concerns about (1) novelty of the proof technique, (2) interpretation of the analysis. The response mitigated some the concerns but did not resolve them. I concur with the reviewers. The paper has some nice insights and good potential. However, there are a few things that need to be clarified and the paper has to be substantially rewritten to reflect this and thus I do not recommend acceptance at this time.